**Glacial to interglacial climate variability in the southeastern African subtropics (25- 20°S)**

Annette Hahn[1], Enno Schefuß[1], Jeroen Groeneveld[1,2], Charlotte Miller[1*], Matthias Zabel[1],

[1]MARUM - Center for Marine Environmental Sciences, University of Bremen, Bremen, Germany

[2]Alfred Wegener Institute, Helmholtz Center for Polar and Marine Research, Potsdam, Germany

*Present address: Leeds Trinity University, Brownberrie Ln, Horsforth, Leeds, LS18 5HD, United
Kingdom

Contact: ahahn@marum.de

Abstract

We present a continuous and well-resolved record of climatic variability for the past 100,000 yrs

from a marine sediment core taken in Delagoa Bight, off southeastern Africa. In addition to

providing a sea surface temperature reconstruction for the past ca. 100,000 yrs, this record also

allows a high-resolution continental climatic reconstruction. Climate sensitive organic proxies, like

the distribution and isotopic composition of plant-wax lipids as well as elemental indicators for

fluvial input and weathering type provide information on climatic changes in the adjacent

catchment areas (Incomati, Matola, and Lusutfu rivers). At the transition between glacials and

interglacials, shifts in vegetation correlate with changes in sea surface temperature in the Agulhas

current. The local hydrology, however, does not follow these orbital-paced shifts. Instead,

precipitation patterns follow millennial scale variations with different forcing mechanisms in

glacial versus interglacial climatic states. During glacials, southward displacement of the

Intertropical Convergence Zone facilitates a transmission of northern hemispheric signals (e.g.

Heinrich events) to the southern hemispheric subtropics. Furthermore, the southern hemispheric

westerlies become a more direct source of precipitation as they shift northward over the study

site, especially during Antarctic cold phases. During interglacials, the observed short-term

hydrological variability is also a function of Antarctic climate variability, however, it is driven by

the indirect influence of the southern hemispheric westerlies and the associated South African

high-pressure cell blocking the South Indian Ocean Convergence Zone related precipitation. As a

consequence of the interplay of these effects, small scale climatic zones exist. We propose a

conceptual model describing latitudinal shifts of these zones along the southeastern African coast

as tropical and temperate climate systems shift over glacial and interglacial cycles. The proposed

model explains some of the apparent contradictions between several paleoclimate records in the

region.

Key words: Delagoa Bight; southern hemisphere westerlies; South Indian Ocean Convergence

Zone; sea surface temperatures; hydrogen isotopes; carbon isotopes; elemental composition

1.      Introduction

Despite the increasing number of southern African paleoclimate studies, large data gaps and

unresolved debates remain. Controversies concern both the interpretation of the climate records

as well as the contradictory major climate forcings that have been proposed for the region. In

southeastern Africa, the main moisture source is the warm Indian Ocean (Tyson and Preston-

Whyte, 2000), the mechanisms controlling the intensity and duration of the easterly rainfall over

time remain, however, uncertain. Climate variations on glacial-interglacial timescales in

southernmost Africa were reported to be directly forced by local (southern hemispheric)

insolation (Partridge et al., 1997; Schefuß et al., 2011; Simon et al., 2015; Caley et al., 2018).

Strong southern hemispheric summer insolation was hypothesized to cause wet climatic

conditions along the east African coast due to a stronger atmospheric convection and an increase

in the land/ocean temperature contrast, which results in higher moisture transport by the tropical

easterlies. However, recent paleo-reconstructions suggested a synchrony with northern

hemisphere climate signals, which are inversely correlated to southern hemispheric insolation

(e.g. Truc et al., 2013). As a mechanism of transmitting the northern hemispheric signal to

southern Africa, ocean circulation variability (Agulhas current strength; i.e. sea surface

temperatures [SST]) has often been proposed (Biastoch et al., 1999; Reason and Rouault, 2005;

Dupont et al., 2011; Tierney et al., 2008; Stager et al., 2011; Scott et al., 2012; Truc et al., 2013;

Baker et al., 2017; Chase et al., 2017). In terms of vegetation shifts, atmospheric $CO_2$ variability

and temperature have been suggested as major driving mechanisms over glacial-interglacial

cycles (Dupont et al., 2019). Nowadays, eastern South Africa is not under the direct influence of

the intertropical convergence zone (ITCZ) as its modern maximum southern extension is ca. 13-

14°S (Gasse et al., 2008). However, the position of the ITCZ was more southerly during glacial

periods (Nicholson and Flohn, 1980; Chiang et al., 2003; Chiang and Bitz, 2005), which may have

allowed ITCZ shifts to reach much further south along the east African coast than today (c.f.
Johnson et al., 2002; Schefuß et al., 2011; Ziegler et al., 2013; Simon et al., 2015). At the same
time, the southern hemispheric westerlies (SHW), which presently influence only the
southernmost tip of Africa, are hypothesized to have moved northward during glacial periods of
increased south Atlantic sea ice extent (Anderson et al., 2009; Sigman et al., 2010; Miller et al.,
2019b). As suggested by Miller et al., (2019b), in such a scenario the temperate systems may have
brought winter moisture to the southeast African coast and/or blocked South Indian Ocean
Convergence Zone (SIOCZ) related precipitation during the summer months. Regional studies
integrating many of the available records have found that; i) several small-scale climatic dipoles
exist due to the interaction of various driving mechanisms and that ii) the spatial extent of these
climatic regions has varied considerably since the last Glacial (Chevalier et al., 2017, Chase et al.,
2018; Miller et al., 2019b). Miller et al., (2019b) compile paleorecords along the southeastern
African coast and propose a conceptual model of climatic variability during the Holocene. The
authors describe three climatic zones; a *northern SRZ* where the climate is driven by local
insolation, and *a central and eastern SRZ* and *southern South African* zone where climate is driven
by shifts of the southern hemisphere westerlies, the South African high-pressure cell and the
SIOCZ. Equatorward shifts of the southern hemisphere westerlies, the South African high-
pressure cell and the SIOCZ result in humid conditions in the *southern South African zone*,
whereas they cause arid conditions in the *central and eastern SRZ.* We analyze a marine core
located within the *central and eastern SRZ* that offers a continuous high-resolution record of the
past ca. 100,000 yrs allowing us to add to the existing conceptual models of southeastern African
climate dynamics, and to gain an understanding of glacial climate mechanisms in the region. A
combination of organic and inorganic geochemical proxies is used in order to decipher the
hydrological processes on land, while foraminiferal shell geochemistry serves as a proxy for ocean
circulation variability. With this approach we aim to decipher some of the discrepancies
concerning the driving mechanisms of southeast African hydroclimate and vegetation shifts
during the last glacial-interglacial cycle.

1.2 Regional setting

The coring site is located in an embayment on the southeastern African continental shelf called
the Delagoa Bight (Fig. 1a). The southern directed Agulhas Current flows along the East African
margin transporting warm and saline water from the tropical Indian Ocean to the tip of

Southern Africa (Zahn et al., 2012). The current system is structured into a series of large-scale

(~200 km diameter) anti-cyclonic eddies occurring about 4 to 5 times per year (Quartly and Sro-

kosz, 2004). As they pass the Delagoa Bight, these eddies, together with the Agulhas Current it-

self, drive the Delagoa Bight eddy; a topographically constrained cyclonic lee eddy at the coring

location (Lutjeharms and Da Silva, 1988; Quartly and Srokosz, 2004). Although the coring site is

located just west of the mouth of the major Limpopo river system, Schüürman et al., (2019)

show that the inorganic material at our site most likely originates from three minor rivers, Inco-

mati, Matola, and Lusutfu, that flow into the Indian Ocean further to the southwest. This is at-

tributed to the eastward deflection of the Limpopo sediments by the Delagoa Bight eddy. The

eddy appears to have been stable and strong enough to effectively constrain the drift of the

Limpopo sediments eastwards over the late Pleistocene and Holocene (Schüürman et al., 2019).

The three rivers, Incomati (also known as Komati), Matola (also known as Umbeluzi), and

Lusutfu (also known as Maputo), have catchment areas of ca. 45 300 km$^2$, 6 600 km$^2$, and 22

700 km$^2$, respectively, comprising the coastal region and the eastern flank of the Drakensberg

Mountains. Between the Drakensberg escarpment and the coast lies a N-S oriented low ridge,

the Lebombo Mountains (400–800 m a.s.l.). The geological formations of this area are the Ar-

chaean Kaapvaal Craton, the Karoo Igneous Province, as well as the Quaternary deposits on the

coastal plains (de Wit et al., 1992; Sweeney et al., 1994). Climatically these catchments are in

the transition zone between tropical and subtropical climate; at the southern limit of the sub-

tropical ridge between the southern Hadley and the Ferrel cell (Tyson and Preston-Whyte,

2000). The average annual temperature ranges from 16°C in the highlands to 24°C in the low-

land area. (Kersberg, 1996). Rain (ca. 1,000 mm annually) falls mostly in summer (ca. 67 % of an-

nual rainfall from November to March) (Xie and Arkin, 1997; Chase and Meadows, 2007). Alt-

hough the ITCZ currently does not directly affect the region, it does induce latitudinal shifts in

the SIOCZ, which can be considered as a southward extension of the ITCZ. When the ITCZ is in

its southernmost (summer) position, tropical temperate troughs (TTTs), forming at the  SIOCZ

bring easterly rainfall from the Indian Ocean (Jury et al., 1993; Reason and Mulenga, 1999) (Fig

1b). During austral summer, a low-pressure cell dominates the Southern African interior, ena-

bling tropical easterlies/TTT to bring rainfall to the region. This rainfall is suppressed during aus-

tral winter, when a subtropical high-pressure cell is located over southern Africa, (Fig. 1b). This

high-pressure cell creates a blocking effect over the continent, which stops moisture advection

inland over the majority of South Africa during winter (Dedekind et al., 2016). The winter rain

that does fall (33 % of annual rainfall from April to October) is associated with extratropical

cloud bands and thunderstorms linked to frontal systems that develop in the main SHW flow

(between 40 °S and 50 °S). As the SHW shift northward during the winter, these frontal systems

may become cut off and displaced equatorward as far north as 25°S (c.f. Baray et al., 2003; Ma-

son and Jury, 1997) (Fig 1c). Associated with this climatological and topographic setting we find

a vegetation in the Incomati, Matola, and Lusutfu catchment areas that consists mainly of

coastal forests and mountain woodlands with savanna elements only in the northernmost parts

of the catchment and sedges along the riverbanks and floodplains (see White, (1983) and

Dupont et al., (2011) for a more detailed description of the vegetation biomes).

2 Material and methods

2.1 Sediments

Gravity core GeoB20616-1 (958 cm long) was retrieved from 25°35.395′S; 33°20.084′E on

15.02.2016 from a water depth of about 460 m. Shipboard sedimentological analysis showed a

lithology of clayey silt with signs of slight bioturbation. The composition was observed as mainly

clastic with occurrence of foraminifera and shell fragments (Zabel, 2016).

2.2 Oxygen isotopic composition of planktonic foraminifera

Stable oxygen isotopes values values of planktonic foraminifera (*G. ruber*, white variety, >150 μm)

were measured in the interval between 395 and 935 cm at 10 cm resolution for age-modeling

(Suppl.1). For each measurement, around eight shells of *G. ruber* were selected and analyzed at

the MARUM – Center for Marine Environmental Sciences, University of Bremen, Germany using

a ThermoFisher Scientific 253 plus gas isotope ratio mass spectrometer with Kiel IV automated

carbonate preparation device. Data were calibrated against an in-house standard (Solnhofen

limestone). The results are reported in permil (‰, parts per thousand) versus Vienna Peedee

belemnite (VPDB). Standard deviation of in-house standard (Solnhofen limestone) $\delta^{18}O$ over the

measurement period was 0.06 ‰.

2.3 Age model

Until the limit of radiocarbon dating the age model used in this study is based on 8 radiocarbon

ages of *G. ruber*, one shell fragment and a bulk total organic carbon surface sample (see Table 1).

The cleaning procedures as well as the Accelerator Mass Spectrometry (AMS) measurements

were carried out in the Poznań Radiocarbon Laboratory, Poland. The modelled ocean average

curve (Marine13) (Reimer et al., 2013) and a local marine ΔR of 121±16 [14]C yr (Maboya et al.,

2017) were applied to calibrate the radiocarbon ages. To perform these calculations the Calib 7.1

software (Stuiver et al., 2019) was used. For flexible Bayesian age-depth modelling of the

available [14]C dates, the software Bacon (Blaauw and Christen, 2011) (Fig. 2b) was used. The

uncertainty of the radiocarbon dates is indicated in Table 1. The uncertainty of the Bacon model

is indicated in Fig. 2b (grey lines). However, there is possibly an underestimation of the error in

the age model around two periods of slow deposition in the interval from 15 to 6 ka BP and in

the interval from 32 to 25 ka BP. The calibrated [14]C age of a shell fragment found in this interval

(390cm) was used as a [14]C-tie-point (see Table 1), additionally 2 $\delta^{18}$O tie-points were defined and

an age model was calculated using the software AnalySeries (Paillard et al., 1996) (Fig. 2a). The

age-depth model was extended by planktonic foraminifera $\delta^{18}$O correlation using major $\delta^{18}$O

shifts in the LR04 stack as a reference (Lisiecki and Raymo, 2005) (Fig. 2a,b). With this low number

of tie-points it is difficult to capture heterogeneity in the deposition rate, which must be

considered when estimating the error of the age model. For the error estimation of $\delta^{18}$O tie-

points the mean resolution of the GeoB20616-1 $\delta^{18}$O record and the reference curve around the

tie-point depth and age (respectively) was taken into account as well as the absolute age error of

the time-scale used for the reference record and a matching error visually estimated when

defining tie-points. Figure 2b (grey lines) gives an estimate of the age model error. In this paper,

we refer to median age estimations.

2.4 Foraminiferal Mg/Ca

Up to 20 specimens (> 150 μm) of *G. ruber* (white) (> 150 μm) were selected for Mg/Ca analysis

(see Suppl.2). Foraminiferal tests were gently crushed prior to standard cleaning procedures for

178  Mg/Ca in foraminifera (Barker et al., 2003). For clay and organic matter removal ultrasonic

cleaning was alternated with washes in deionized water and methanol, an oxidizing step with 1

180  %-$H_2O_2$ buffered in 0.1M NaOH followed, which was then neutralized by deionized water washes.

A final weak acid leach with 0.001M QD $HNO_3$ was performed before dissolution in 0.5 mL

0.075 M QD $HNO_3$ and centrifugation for 10 min (6,000 rpm). The samples were diluted with

Seralpur water before analysis with inductively coupled plasma optical emission spectrometry

(Agilent Technologies, 700 Series with autosampler ASX-520 CETAC and micro-nebulizer) at

MARUM, University of Bremen, Germany. Instrumental precision was monitored after every five

samples using analysis of an in-house standard solution with a Mg/Ca of 2.93 mmol $mol^{-1}$

(standard deviation of 0.020 mmol $mol^{-1}$ or 0.67 %). A limestone standard (ECRM752-1, reported

188 Mg/Ca of 3.75 mmol $mol^{-1}$) was analyzed to allow inter-laboratory comparison (Greaves et al.,

2008; Groeneveld and Filipsson, 2013).

2.5 Organic geochemistry

Total lipid extracts (TLEs) were extracted from ca. 9-27 g of the freeze-dried, homogenized

samples with a DIONEX Accelerated Solvent Extractor (ASE 200) at 100°C and at 1,000 psi for 5

minutes (repeated 3 times) using a dichloromethane (DCM):methanol (MeOH) (9:1, v/v) mixture.

Squalane was added in a known amount to the samples as internal standard before extraction.

Elemental sulphur was removed from the TLEs using copper turnings. After saponification by

adding 6 % KOH in MeOH and extraction of the neutral fractions with hexane, the neutral fractions

were split into hydrocarbon, ketone, and polar fractions using silica gel column chromatography

(with a mesh size of 60 μm) and elution with hexane, DCM and DCM:MeOH (1:1), respectively.

Subsequently elution of the hydrocarbon fractions with hexane over an $AgNO_3$-impregnated silica

column yielded saturated hydrocarbon fractions. The concentrations of long-chain $n$-alkanes in

the saturated hydrocarbon fractions were determined using a Thermo Fischer Scientific Focus

gas-chromatograph (GC) with flame-ionization-detection (FID) equipped with a Restek Rxi 5ms

column (30m x 0.25mm x 0.25μm). Quantities of individual $n$-alkanes were estimated by

comparison with an external standard containing $n$-alkanes ($C_{19}$–$C_{34}$) at a known concentration.

Replicate analyses of the external standard yielded a quantification uncertainty of <5 %. The

carbon preference index (CPI) was calculated using the following equation:

CPI = 0.5 * ($\sum C_{odd27-33}$/ $\sum C_{even26-32}$ + $\sum C_{odd27-33}$/ $\sum C_{even28-34}$) with $C_x$ the amount of each

homologue (Bray and Evans 1961).

The δD values of long-chain $n$-alkanes were measured using a Thermo Trace GC equipped with an

Agilent DB-5MS (30m length, 0.25 mm ID, 1.00 μm film) coupled via a pyrolysis reactor (operated

at 1420°C) to a Thermo Fisher MAT 253 isotope ratio mass spectrometer (GC/IR-MS). The δD

values were calibrated against external $H_2$ reference gas. The $H^{3+}$ factor was monitored daily and

varied around $6.23 \pm 0.04$ ppm $nA^{-1}$. δD values are reported in permil (‰) versus Vienna Standard

Mean Ocean Water (VSMOW). An *n*-alkane standard of 16 externally calibrated alkanes was

measured every 6[th] measurement. Long-term precision and accuracy of the external alkane

standard were 3 and <1 ‰, respectively. When *n*-alkane concentrations permitted, samples were

run at least in duplicate. Precision and accuracy of the squalane internal standard were 2 and <1

218 ‰, respectively (n=41). Average precision of the *n*-$C_{29}$ alkane in replicates was 4 ‰. The $\delta^{13}C$

values of the long-chain *n*-alkanes were measured using a Thermo Trace GC Ultra coupled to a

Finnigan MAT 252 isotope ratio monitoring mass spectrometer via a combustion interface

operated at 1,000°C. The $\delta^{13}C$ values were calibrated against external $CO_2$ reference gas. $\delta^{13}C$

values are reported in permil (‰) against Vienna Pee Dee Belemnite (VPDB). When

concentrations permitted, samples were run at least in duplicate. Precision and accuracy of the

squalane internal standard were 0.1 and 0.4 ‰, respectively (n=41). An external standard mixture

was analyzed repeatedly every 6 runs and yielded a long-term mean standard deviation of 0.2 ‰

with a mean deviation of 0.1 ‰ from the reference values. Average precision of the *n*-$C_{29}$ alkane

in replicates was 0.3 ‰. We focus the discussion on the isotopic signals of the *n* -$C_{31}$ alkane, as

this compound is derived from grasses and trees present throughout the study area. Supplement

3 shows, however, that the *n* -$C_{29}$ and *n* -$C_{33}$ alkanes reveal similar trends.

2.6 Inorganic geochemistry

The elemental composition of all onshore and offshore samples was measured using a

combination of high resolution (1 cm) semi-quantitative XRF scanning and lower (5 cm) resolution

quantitative XRF measurements on discrete samples (see Suppl. 4). XRF core scanning (Avaatech

XRF Scanner II at MARUM, University of Bremen) was performed with an excitation potential of

10 kV, a current of 250 mA, and 30 s counting time for Ca, Fe, K and Al. For discrete measurements

on 110 dried and ground samples, a PANalytical Epsilon3-XL XRF spectrometer equipped with a

rhodium tube, several filters, and a SSD5 detector was used. A calibration based on certified

standard materials (e.g. GBW07309, GBW07316, and MAG-1) was used to quantify elemental

counts (c.f. Govin et al., 2012).

3 Results and discussion

## 3.1 Proxy indicators

### 3.1.1 SST

The magnitude of temperature variability (from ca. 27°C during interglacials to ca. 24°C during glacials) in the GeoB20616-1 Mg/Ca SST record and the timing of changes (postglacial warming at ca. 17 ka BP) correspond to existing regional Mg/Ca SST records (c.f. Fig. 3; Bard et al., 1997; Levi et al., 2007; Wang et al., 2013). They do, however, not correspond to SST calculated from other indicators (i.e. $U^{K'}_{37}$, $TEX^{86}$) (e.g. Wang et al., 2013; Caley et al., 2011). These indicators show slightly different patterns, which may be attributed to a seasonal bias in the proxies (Wang et al., 2013). Wang et al., (2013) suggest that $U^{K'}_{37}$ SST reflects warm season SST mediated by changes in the Atlantic, whereas the *G. ruber* Mg/Ca SST indicator used in this study records cold season SST mediated by climate changes in the southern hemisphere.

### 3.1.2 Vegetation signatures

The $\delta^{13}C_{wax}$ record of core GeoB20616-1 shows average values of approximately -24‰ VPDB (c.f. Suppl. 3) and shifts from ca. -25 ‰ to ca. -24 ‰ (at 85 ka BP) and from -24 ‰ to − 25 ‰ (at ca. 10 ka BP). The stable carbon isotopic composition of plant waxes reflects discrimination between $^{12}C$ and $^{13}C$ during biosynthesis varying with vegetation type: $C_4$ plants have higher $\delta^{13}C$ values than $C_3$ plants (e.g., Collister et al., 1994; Herrmann et al., 2016). The average $\delta^{13}C$ value of the analyzed samples falls into the range between $C_3$ alkanes (around -35‰) and $C_4$ alkanes (around -20‰) (Garcin et al., 2014) indicating that the *n*-alkanes were derived from $C_3$ sources in the catchment such as mountain shrublands and coastal forests, as well as from $C_4$ sedges which grow along rivers and in the associated swamplands (c.f. Fig. 1a). There is no correlation ($R^2$=0.15) of $\delta^{13}C_{wax}$ variability and hydrological variability indicated by $\delta D_{wax}$ (see section 3.1.3 Precipitation indicators for details on this proxy). We therefore suggest that the shifts we see in the $\delta^{13}C_{wax}$ werenot induced by a xeric/mesic adaptation of the same plant community. Instead, we imply that the shifts in the $\delta^{13}C_{wax}$ signal were related to shifts in the vegetation community. Palynological work on a nearby marine sediment core by Dupont et al. (2011) shows that large shifts in vegetation biomes are also observed in the Limpopo catchment which is directly adjacent to the Incomati, Matola and Lusutfu catchments (Fig. 1a). A comparison of the Dupont et al. (2011) palynological data (Fig. 3c) and the $\delta^{13}C_{wax}$ data at our site (Fig. 3a) shows a covariation of

major shifts in vegetation and $\delta^{13}C_{wax}$. Although the similarities in the pattern of vegetation shifts detected in the nearby Limpopo river sediment core and at our study site suggest that large scale vegetation shifts took place in the region over glacial – interglacial transitions, this does not necessarily imply the mechanisms behind these trends are the same. Studies of the Limpopo sediment record (Dupont et al. 2011; Caley et al. 2018) reveal a $\delta^{13}C_{wax}$-enriched grassland vegetation for glacial intervals and an increase of woodland vegetation during well-developed interglacial periods, as is the case for MIS 5 and 1 (as opposed to MIS 3), reflected in lighter $\delta^{13}C_{wax}$ values. Caley et al., (2018) attribute the $\delta^{13}C_{wax}$-enrichment in Limpopo river sediments during glacials to an expansion of floodplains and the associated $C_4$ sedges, as well as discharge from the upper Limpopo catchment which reached well into the grassland interior of southern Africa (almost 1,000 km inland). The headwaters of the Incomati, Matola, and Lusutfu catchment areas, however, are in the Lebombo mountain range located within 200 km of the coast. They do not reach into the interior grassland biomes of South Africa. We therefore propose that in the Incomati, Matola, and Lusutfu catchment areas, the heavier $\delta^{13}C_{wax}$ values for the glacial MIS 4-2 interval reflect retreating forests and an expansion of drought tolerant $C_4$ plants (grasses) due to growing season aridity, whereas interglacial (MIS 1 and 5) lighter $\delta^{13}C_{wax}$ values reflect the formation of woodlands. Furthermore, sedge-dominated open swamps that fringed rivers during MIS 4-2 may have been replaced by gallery forests during MIS1 and 5 contributing to the glacial to interglacial $\delta^{13}C_{wax}$ depletion.

3.1.3 Precipitation indicators

Hydrogen isotope changes measured in plant waxes are related to the isotope composition of precipitation since hydrogen used for biosynthesis originates directly from the water taken up by the plants (Sessions et al., 1999). In tropical and subtropical areas, the isotopic composition of rainfall (δDp) mainly reflects the amount of precipitation - with δDp depletion indicating more rainfall (Dansgaard, 1964). Furthermore, rainfall δDp signatures may also become deuterium-depleted with altitude (ca. 10–15 ‰ per 1,000 m, Gonfiantini et al., (2001)). The δD values of leaf waxes in the three catchments are probably affected by both the amount as well as the altitude effect. Rainfall at higher altitudes takes place during times of generally increased rainfall, as it is high precipitation events that reach the interior. The altitude effect therefore enhances the δD depletion of the "amount effect". The K/Al ratio of the sediment is a less direct indicator of the precipitation regime: K/Al has been interpreted as an index between illite $(K,H_3O)$ and kaolinite

$(Al_2Si_2O_5(OH)_4)$ giving an indication of the prevailing weathering regime as illite is a product of

physical weathering whereas kaolinite is produced during chemical weathering (Clift et al., 2008;

Dickson et al., 2010; Burnett et al., 2011). The Ca/Fe ratio is generally used as a proxy of marine

(Ca) versus terrestrial (Fe) input to the core site and thus indicative of changes in terrestrial

discharge by the river systems (Hebbeln and Cortés, 2001; Croudace et al., 2006; Rogerson et al.,

2006; Rothwell and Rack, 2006; McGregor et al., 2009; Dickson et al., 2010; Nizou et al., 2010).

The red/blue ratio of the sediment reflects sediment color nuance and increases with sediment

lightness. In Geob20616-1 we interpret the reddish values as a more clastic deposition indicative

of arid conditions whereas darker blueish colors may reflect clay and organic rich sediments

preferentially deposited during humid phases (see also M123 cruise report Zabel, 2006). In the

records of $\delta D_{C31}$, red/blue , K/Al and Ca/Fe similar patterns can be observed: They all display

relatively high values (up to -144 ‰, 1.4; 12 and 0.25 respectively) in the intervals marked in

red/yellow in Fig.4 and lower values (down to -160, 1.1, 1, and 0.2 respectively) in the intervals

marked in blue/green in Fig. 4. We associate these variations with (respectively) decreasing

(red/yellow) and increasing (blue/green) precipitation over the Incomati, Matola, and Lusutfu

catchment areas. We note that the observed correlation, in particular for the inorganic proxies

(K/Al and Ca/Fe), is relative rather than absolute in nature. This can be associated with the

changing background conditions over glacial and interglacial cycles which may cause shifts in the

elemental composition. We also note that of the four proxy indicators ($\delta D_{C31}$, red/blue, K/Al and

Ca/Fe) only $\delta D_{C31}$ can be considered as direct indicator of past precipitation change. Red/blue,

K/Al and Ca/Fe depend to varying extents on precipitation, erosion and fluvial transport, whereas

these factors do not necessarily vary in concert. For instance, erosion is not always directly linked

to the amount of precipitation and vegetation density is often an additional and more important

factor for erosion rates. Erosion rates can also increase substantially at times of rapid climatic and

associated vegetation changes. Because the relationship between precipitation, erosion and

riverine transport is not linear we base our precipitation reconstruction (i.e. the definition of the

arid and wet intervals described in section 3.2 and colored-coded in Fig. 4) mainly on the $\delta D_{C31}$

values. We consider the red/blue, K/Al and Ca/Fe values as supportive information; the relative

correlation of the four proxies suggests that phases of increased precipitation are, for the most

part, associated with an increase in erosion rates, chemical weathering and riverine transport.

This underlines the reliability of our paleo-precipitation reconstruction.

### 3.2 Climatic patterns at different time scales

### 3.2.1 Orbital time scales

*3.2.1.1. Sea surface temperatures and vegetation*

Over the past 100,000 yrs the SST and $\delta^{13}C_{C31}$ values show a common trend of high SST and low $\delta^{13}C_{wax}$ values during interglacial MIS 5 and 1 and low SST and high $\delta^{13}C$ values during glacial MIS 4-2 (Fig. 3). Our data reveal an increase in SST of ca. 4°C from glacial to interglacial conditions. This correlation between SST and glacial-interglacial changes cycles is commonly found for this area (Caley et al., 2011; Dupont et al., 2011; Caley et al., 2018). On this glacial-interglacial time scale, variations in local SST are thought to be an important driver of hydroclimate in southeastern Africa (c.f. Dupont et al., 2011). During interglacials, warm SST within the Mozambique Channel and Agulhas Current induce an advection of moist air and higher rainfall in the east South African summer rainfall zone (e.g. Walker, 1990; Reason and Mulenga, 1999; Tyson and Preston-Whyte, 2000). The opposite effect is inferred for glacial periods (Dupont et al., 2011; Chevalier and Chase, 2015). The strong influence of western Indian Ocean surface temperatures on the summer precipitation in northern South Africa and southern Mozambique induces a tight coupling between vegetation dynamics in southeastern Africa and sea surface temperature variations in the Western Indian Ocean. This has been shown for several glacial – interglacial cycles in a palynological study offshore Limpopo River (core MD96-2048; Fig. 1a) by Dupont et al., (2011).

*3.2.1.2. Hydrology over glacial-interglacial transitions*

$\delta D$, XRF, and color data are indicators of catchment precipitation changes: decreases in red/blue, Ca/Fe, K/Al ratios and $\delta D$ values indicate higher precipitation in the catchment, more fluvial discharge and higher chemical weathering rates (see section 3.1.3). Although there is much variability in the hydrological record of core GeoB20616-1, red/blue, Ca/Fe, K/Al ratios and $\delta D$ values are surprisingly stable over glacial –interglacial transitions (mean $\delta D$ value of MIS 1 and 5: -149 ‰ versus mean $\delta D$ value of MIS 2-4: -150 ‰). It can be assumed that, during glacials, the rainfall from the main rain bearing systems (SIOCZ related tropical temperate troughs) was reduced due to generally lower land- and sea-surface temperatures and a weaker global hydrological cycle. However, a southward shift of the ITCZ during glacials as previously suggested

(Nicholson and Flohn, 1980; Johnson et al., 2002; Chiang et al., 2003; Chiang and Bitz, 2005;
Schefuß et al., 2011) would have contributed to increased rainfall in the study area. It is unclear
if the region would have been under the direct influence of the ITCZ during glacials or if southward
shifts of the ITCZ entailed a southward shift of the SIOCZ and thus increased precipitation via the
TTT. Furthermore, SHW related low pressure systems shifting northward to the Incomati, Matola
and Lusutfu catchment areas during glacial conditions may have become a major additional
precipitation source. The SHW northward shift of ca. 5° latitude is well documented (Chase and
Meadows, 2007; Chevalier and Chase, 2015; Chase et al., 2017; Miller et al., 2019a). The
possibility of more frequent SHW related low pressure systems bringing moisture to our study
area during the LGM has previous been proposed by Scott et al., (2012) in the framework of a
regional pollen review paper. It is also suggested by a modelling study showing an LGM scenario
of drier summers and wetter winters for the southeastern African coast (Engelbrecht et al., 2019).
During glacial periods, a reduced summer (SIOCZ related) rainfall amount and an increase in SHW
related frontal systems as an additional winter precipitation source, possibly in combination with
precipitation from a more southerly ITCZ, would translate to a relatively stable annual rainfall
amount over glacial-interglacial transitions.

3.2.2 Millennial scale hydrological variability

*3.2.2.1 During Interglacial MIS 5*

During MIS 5 there are several prominent (ca. -10 ‰) short-term (1-2 ka) decreases in the δD
record, which are paralleled with decreases in Ca/Fe, K/Al and red/blue ratios (Fig. 4). We
interpret these intervals (approximately 83-80 ka BP and 93-90 ka BP) as wet periods while
intervals of high Ca/Fe, K/Al and red/blue ratios and δD values (approximately 97-95 ka BP, 87.5-
85 ka BP and 77.5 ka BP) are interpreted as arid intervals (see section 3.1.2. for details on proxy
interpretation). During the interglacial MIS 5, millennial scale increases in humidity correlate
broadly to periods of warmth in the Antarctic ice core records termed AIM22 and AIM 21 (AIM:
Antarctic isotope maxima) (see Fig. 4; EPICA members, 2010). During these Antarctic warm
periods, sea ice, the circumpolar circulation and the SHW retracted. This is recorded by Southern
Ocean diatom burial rates as well as paleoclimate archives at the southernmost tips of Africa and
South America (Lamy et al., 2001; Anderson et al., 2009; Chase et al., 2009; Hahn et al 2016 and
references therein; Zhao et al., 2016). It has been hypothesized that southward shifts of the SHW

and the South African high-pressure cell, allow the SIOCZ and TTT to shift further south causing an increase in humidity in our study area. Miller et al., (2019b) suggest this mechanism for the region just south of our site (termed *eastern central zone*), which shows Holocene hydroclimatic shifts similar to those recorded in GeoB20616-1. Holocene arid events in this region are attributed to northward shifts of the SHW and the South African high-pressure cell which block the SIOCZ and TTT related moisture. These mechanisms are described in detail by Miller et al., 2019b and our data suggests that they were also active during earlier interglacial periods (e.g. MIS 5) (c.f. schematic model in Fig. 5a). Our current chronology suggests that southward SHW shifts during Antarctic warm periods caused the prominent humid phases during MIS 5 in the Incomati, Matola and Lusutfu catchment areas during the timeframes around 83-80 ka BP (AIM21) and 93-90 ka BP (AIM22). When our best age estimate is applied there is little correspondence between northern or southern insolation maxima and the MIS5 humid phases. In view of the chronological uncertainty in this early part of the record (beyond the [14]C dating limit), we cannot exclude that these humid phases are related to precessional variability, in the absence of ice interference, causing the division in MIS5a-e. However, in accordance with the conceptual model by Miller et al., (2019b) for the Holocene, we observe no local insolation control on climate at our study site. We suggest that the major shifts in the large-scale rain-bearing systems may override the local insolation forcing.

*3.2.2.2. During MIS 4-2 glacial conditions*

During the glacial periods MIS 2 and 4 and the less prominent interglacial MIS 3, the correlation between southeastern African humidity and Antarctic warm periods (AIM events) does not persist. In contrast; the first two prominent humid phases in MIS 4 (around 68-63 ka BP and 56 ka BP) as well as some of the following more short-term humid phases coincide with cold periods in the Antarctic ice core record (Fig.4). The general position of the SHW trajectories is suggested to have been located 5° in latitude further north during glacial periods (c.f. section *3.2.1.2. Hydrology over glacial-interglacial transitions*). The Incomati, Matola and Lusutfu catchment areas would therefore have been in the direct trajectory of the SHW related low pressure systems. Whilst northward shifts of the SHW and the South African high pressure cell during an interglacial cause aridity by blocking the SIOCZ and TTT (as suggested by Miller et al.,(2019b) and as described in section 3.2.2.1 for e.g. MIS 5), we suggest that during a glacial, additional northward shifts of the SHW (e.g. during Antarctic cold events) would have led to an increase in precipitation related

to particularly strong direct influence of the SHW and the related low pressure cells (c.f. schematic model Fig 5b). Fig. 4 also shows a correlation between some of the humid phases during MIS 2-4 and Greenland cold phases i.e. Heinrich stadials. The timing of the wet phases at 68-63 ka, 56 ka, 44 ka, 37 ka,  and 23 ka BP corresponds roughly to the following Heinrich stadials: HS6 (after 60 ka BP, Rasmussen et al., 2014); HS5a (56 ka BP, Chapman and Shackleton, 1999); HS5 (45 ka BP; Hemming 2004) and HS4 & HS2 (37 ka BP and 23 ka BP, Bond and Lotti, 1995). Wet phases in eastern Africa have previously been associated with Heinrich events (Caley et al., 2018; Dupont et al., 2011; Schefuß et al., 2011). It is well documented that during glacial conditions the large ice masses of the northern hemisphere displace the thermal equator southward (Nicholson and Flohn, 1980; Johnson et al., 2002; Chiang et al., 2003; Chiang and Bitz, 2005; Schefuß et al., 2011). It is therefore hypothesized that the ITCZ reached latitudes further south than its modern maximal extent causing the MIS 2-4 rainfall peaks. There is no notable "blocking" effect of the South African high-pressure cell during glacials (schematic model Fig. 5b). The transitions from cold "stadial" to warm "interstadial" conditions and back during MIS 2-4 are extremely rapid and short term. The sampling resolution and age – control of our record (especially prior to ca. 50 ka BP – the limit of $^{14}$C dating) is not always sufficient for capturing these variations (e.g. HS4). The association of humid phases with a northward shifting SHW and/or southward shifting ITCZ is therefore not always clear and a combination of both may also be possible.

*3.2.1.3 From the LGM to the Holocene*

Relative to the prolonged arid phase during the late MIS 3/early MIS 2 (37-25 ka BP; c.f. Fig. 4), we observe a trend towards more humid conditions during the LGM (25 – 18 ka BP) marked by a decrease in Ca/Fe, K/Al, red/blue ratios and δD values. This is most likely due to the more frequent SHW-related low-pressure systems bringing moisture to our study area during the LGM and/or southward shifts of the ITCZ as discussed in section *3.2.1.2. Hydrology over glacial-interglacial transitions* (see also Fig. 5b). Our record shows a wettening trend after the Last Glacial Maximum and during the deglacial (from ca. 15 ka BP). Several paleoenvironmental records show a common humidity increase for this interval (Meadows 1988; Scott 1989; Norström et al., 2009). Chase et al.,(2017) attribute this to the invigoration of tropical systems with post-glacial warming. The wet conditions prevail until the early Holocene (ca. 8 ka BP). Similar observations of a ca. 15-8 ka BP wet phase have been made in the region (e.g. Norström et al., 2009; Neumann et al., 2010). For this early -Mid Holocene period, we infer from the leaf wax δ$^{13}$C values a shift from grassland to

453 woodlands as described in section 3.2.1.1. and in Dupont et al. (2011). This may be related to the

454 rainfall intensification as well as to the global temperature and $CO_2$ increase (c.f. Dupont et al.,

2019). The early/Mid Holocene wet phase in our study region (*eastern central SRZ*) is described

by Miller et al., 2019b and associated with a southward shift of the SHW and the South African

high-pressure cell allowing for the SIOCZ related rain bearing systems (TTT) to shift southward

over the region. The late Holocene (the last 5 kyrs) however, was an arid phase at our study cite

as suggested by the precipitation indicators δD, Ca/Fe, K/Al and red/blue ratios. Several regional

records (e.g. Mfabeni peatlands and the *eastern-central region*) show similar shifts; from a wet

deglacial / Early Holocene (18-5 ka BP) to dry conditions thereafter (Chevalier et al., 2015; Miller

et al., 2019a). Miller et al. (2019b) compile eastern African climate records and recognize a late

Holocene tripole of increased humidity north of 20°S and south of 25°S and a contrasting aridity

trend in the region in-between. Our catchment is located at the northernmost extent of this

intermediate region; while we record an aridity trend in the Late Holocene, the adjacent Limpopo

catchment just to the north received higher rainfall amounts during this time interval (Miller et

al., 2019b). A northward shift in SHW with the South African high-pressure cell blocking the SIOCZ

and TTT is a suggested mechanism for this late Holocene aridity (Miller et al., 2019b; also

described in section 3.2.2.1). Likewise, Mason and Jury (1997) (based on a conceptual model by

Tyson (1984)) suggest that northward shifting SHW induce rain-bearing low pressure cells to shift

away from the eastern African coast towards Madagascar. During the Late Holocene the modern

climatic situation of the study area was established: during the summer months the SHW and the

South African high-pressure cell are in their southernmost position allowing the SIOCZ related

TTT to bring rainfall to the region (66 % of annual precipitation). During the winter months the

SHW and the South African high-pressure cell shift northward. In this constellation the SIOCZ and

TTT influence are blocked by the South African high-pressure cell, however low-pressure cells

may become cut from the main SHW flow bringing winter rainfall to the area (33 % of annual

precipitation) as described in section 1.2.

Conclusions

Using the organic and inorganic geochemical properties of sediment core GeoB20616-1 from the

Delagoa Bight we were able to reconstruct the vegetation changes and rainfall patterns in the

Incomati, Matola and Lusutfu catchments as well as SST trends of the Agulhas waters for the past

ca. 100,000 yrs offshore southeastern Africa. Our reconstructions underline the existing dipoles

or tripoles in southeastern African climate: although the glacial-interglacial variability at our site resembles that observed in the adjacent Limpopo river catchment, the Holocene hydrological trends are exactly inverted in these neighboring catchments. Small-scale climatic zones have been previously described for the region (c.f. Scott et al., 2012; Chevalier and Chase, 2015; Miller et al., 2019b) and each zone has been attributed to a climatic driving mechanism. Our data provide insights into the spatial shifts of these zones as fundamental shifts in the major climate systems occurred over glacial-interglacial cycles. In accordance with Miller et al., (2019b) we identify displacements of the SHW as the main hydro-climate driver during the Holocene in our study area (termed *central and eastern zone*). The main trajectories of the SHW related disturbances remain so far south during the Holocene, that they rarely deliver direct rainfall to the study area. Instead, northward shifts of the SHW and the South African high-pressure cell block the SIOCZ and thus TTT related rainfalls over the region (Fig. 5a). In this manner latitudinal SHW shifts influence the local rainfall indirectly. Our study not only confirms the Miller et al. (2019 b) conceptual model for the Holocene, but also finds the same mechanisms to be active during MIS5. Similar to Miller et al. (2019b) we find an absence of insolation forcing in our study area. We suggest that at these latitudes local insolation as a climatic forcing mechanism is overridden by shifts in the major rain-bearing systems. We conclude that during interglacials regional wet phases are induced by southward shifting westerlies (related to Antarctic warming trends) allowing for the influence of the SIOCZ related TTT. During glacial periods, however, we observe an inverted relationship between Antarctic warm events and regional humidity, and an additional correlation of several humid intervals with extreme northern hemispheric cold events (HS). This suggests that the mechanisms driving the millennial scale hydrological variability during glacials are not the same as during interglacials. We attribute this to the global reorganization of climate systems during the glacial as the large ice masses at both poles induce a southward shift of the thermal equator and the ITCZ as well as a northward shift of the SHW. Our study site is located at the interface of these "compressed" climate systems. As a result, during full glacial conditions, the region may have received precipitation both from SHW related disturbances as well as from SIOCZ related TTT (Fig. 5b). In this "compressed" state the northward shifts of the SHW and the South African high pressure no longer have the net effect of blocking SIOCZ related precipitation; as this is compensated by the increase in winter rains. Overall humidity therefore shows no considerable decrease during MIS 2-4. Nevertheless, a shift in vegetation from woodland to grasslands takes place during glacials; we attribute this to a reduced growing-season (summer) precipitation,

probably in combination with low temperatures and atmospheric $CO_2$. Our study shows that

these mechanisms are active in a spatially very restrained area resulting in small-scale variability.

These small-scale climatic dipoles or tripoles make the southeastern African coastal area

especially sensitive to shifts in the global climatic system.

Acknowledgments

This work was financially supported by Bundesministerium für Bildung und Forschung (BMBF,

Bonn, Germany) within the projects "Regional Archives for Integrated Investigation (RAiN),"

project number: 03G0840A and "Tracing Human and Climate impacts in South Africa (TRACES)"

project number: 03F0798C. The captain, crew, and scientists of the Meteor M123 cruise are

acknowledged for facilitating the recovery of the studied material. This study would not have

been possible without the MARUM—Center for Marine Environmental Sciences, University of

Bremen, Germany and the laboratory help of Dr. Henning Kuhnert, Ralph Kreutz and Silvana Pape.

In particular, we thank the GeoB Core Repository at the MARUM and Pangaea (www.pangaea.de)

for archiving the sediments and the data used in this paper. Thanks to all RAiN members as well

as Stephan Woodborne and the anonymous reviewer of this manuscript for critical comments

and helpful advice.

Captions

Fig. 1 A: Modern vegetation of southern Africa and the Incomati, Matola and Lusutfu catchments (after White 1983) and annual SST over the Indian Ocean (Locarnini et al., 2013). Grey arrows represent the main easterly transport of moisture from the warm Indian Ocean. The Mozambique current (MC), Agulhas current (AC), and counter current (cc) forming a coastal eddy are shown in black. Sites mentioned in the discussion are numbered as: 1) Wonderkrater (Truc et al., 2013); 2) Braamhoek (Norström et al., 2009); 3) Mfabeni (Miller et al., 2019a); 4) MD96-2048 (Dupont et al., 2011; Caley et al., 2011, 2018); 5) GeoB20610-1 (Miller et al., 2019b); 6) GIK16160-3 (Wang et al., 2013); 7) MD79-257 (Bard et al., 1997; Sonzogni et al., 1998; Levi et al., 2007); 8) GeoB9307-3 (Schefuß et al., 2011). B: Map of South Africa in austral summer showing the shematic postion of the low-pressure system, the ITCZ (Intertopical Convergence Zone), the SIOCZ (South Indian Ocean convergence zone) and related rain bearing TTT (tropical temperate troughs). C: Map of South Africa in austral winter showing the shematic postion of the high-pressure system, the

weaker TTT (tropical temperate troughs) and the frontal systems associated with the northward shifted SHW (southern hemispheric westerlies).

Fig. 2 Reference curves and age–depth model of core GeoB20616-1. A: LR04 benthic foraminifera $\delta^{18}O$ stack (Lisiecki and Raymo, 2005) (black) compared to GeoB20616-1 (red) *G. ruber* foraminifera $\delta^{18}O$ with indicated tie points. B: Age-depth model based on Bacon v. 2.2 (Blaauw and Christen, 2011; green) and $\delta^{18}O$ correlation (blue). Blue circles in panel B represent the positions of calibrated $^{14}C$ ages whereas blue circles indicate $\delta^{18}O$ tie points. Grey lines indicate uncertainty.

Fig. 3 Climatic patterns at orbital time scales recorded in GeoB20616-1. Panel a) shows down-core $\delta^{13}C$ values of the $C_{31}$ *n*- alkane in ‰ VPDB of GeoB20616-1 as indicators for shifts in vegetation type ($C_3$ vs. $C_4$). Panel b) shows SST (sea surface temperatures) recorded by *G. ruber* Mg/Ca (black line) in GeoB20616-1 as well as offshore Limpopo River (core MD96-2048) SST calculated from $TEX_{86}$ (dashed line) and from $U^{K'}_{37}$ (grey line) (Caley et al., 2011). Panel c) shows Limpopo vegetation endmember EM2 from Dupont et al. (2011). The diamonds indicate $C^{14}$ dates (red) and $\delta^{18}O$ tie points (orange).

Fig. 4 Millennial scale hydrological variability recorded in core GeoB20616-1. Organic and inorganic down-core geochemistry (c-f: δD, red/blue, K/Al and Ca/Fe) of GeoB20616-1 as indicators for weathering type, fluvial input and aridity. Intervals identified as wet using these indicators are marked in blue or green, while dry phases are marked in red or yellow. Wet intervals marked in green are associated with southward shifts of the SHW (southern hemispheric westerlies) and the South African high-pressure cell allowing for the SIOCZ (South Indian Ocean convergence zone) and related rain bearing TTT (tropical temperate troughs) to move over the study area during interglacials. In turn, wet intervals marked in blue are associated with northward shifts of the SHW and/or southward shifts of the ITCZ during glacials. Arid phases during interglacials (marked in yellow) are related to northward shifts of the SHW as this induces the moisture-blocking effect of the South African high-pressure cell over the region. During glacials, however, southward shifts of the SHW are often associated with arid phases (marked in red) as the rain-bearing systems related to the SHW move south. Transitional intervals between arid and wet intervals are not colored. XRF scanning data is marked as a line, whereas discrete XRF measurements are represented by points. Panel c represents the δD of the $C_{31}$ *n* alkane in

the unit ‰ VSMOW. For comparative purposes local insolation (Laskar, 2011) as well as Arctic and Antarctic ice core d$^{18}$O records are plotted (NGRIP members, 2004; EPICA members, 2010). The most prominent AIM (Antarctic isotope maxima) and HS (Heinrich Stadial) events are named. The diamonds indicate C$^{14}$ dates (red) and δ$^{18}$O tie points (orange).

Fig. 5 Conceptual model of precipitation shifts during glacial vs. interglacial (present conditions) intervals. The blue shaded boxes indicate the locations of the major regional rain-bearing systems: i) the TTT (tropical temperate troughs) moisture shifting with the SIOCZ (south Indian Ocean convergence zone) and bearing summer rain (therefore marked as SR) ii) the low-pressure systems related to the SHW (southern hemispheric westerlies), bringing mainly winter rain (therefore marked as WR). The orange shaded box marks South African high-pressure cell (HPC) shifting with the SHW. The HPC blocks SIOCZ and TTT related moisture and therefore causes aridity. The arrows mark the millennial scale variability of the position of these systems over the study area which is marked by a star. Please note that the millennial scale variations that the region experiences differ in the interglacial state (box A) and the glacial state (box B) since the organization of the major climatic systems (marked in red) is different ("decompressed" vs "compressed"). The conceptualization for interglacial states presented in box A is based on a schematic model by (Miller et al., 2019b). In this "decompressed" state latitudinal shifts of the SHW indirectly control precipitation at our study site via the moisture blocking effect of the South African HPC: southward shifts of the SHW and HPC allow the SIOCZ related TTT to bring SR to our site, whereas northward movements block this SR moisture (Miller et al., 2019b). During the "compressed" glacial state (box B) the SHW related WR reaches much further north, directly influencing the study site. The SR, in turn is shifted southward and an HPC blocking effect is not noted at our site.

Table. 1 AMS radiocarbon analyses of material from core GeoB20616-1. The modelled ocean average curve (Marine13) (Reimer et al., 2013) was used for calibration and a local ΔR of 121±16 $^{14}$C yr (Maboya et al., 2017) was applied. The ages were calibrated with Calib 7.1 software (Stuiver et al., 2019)

Supplement 1 GeoB20616-1 Oxygen and carbon isotopic composition of planktonic foraminifera (*G.ruber*).

Supplement 2 GeoB20616-1 downcore sea surface temperatures (SST) calculated following Lea et al., 2003 using Mg/Ca analysed on the planktonic foraminifer *G. ruber* (in mmol/mol).

Supplement 3 GeoB20616-1 organic geochemical down-core data. *n*-alkane isotopic composition and distribution descriptive parameters averaged. The elevated CPI values ranging from 3.8 to 14 indicate that the *n*-alkanes within the terrestrial and marine samples were likely derived from non-degraded, terrestrial, higher plant material (Eglinton & Hamilton, 1967). We focus the discussion on the isotopic signals of the *n*-$C_{31}$ alkane but note that the *n*-$C_{29}$ and *n*-$C_{33}$ alkanes reveal similar trends.

Supplement 4 GeoB20616-1 inorganic geochemical down-core data from discrete XRF measurements.

Contributor Roles

Annette Hahn: conceptualization, investigation, analysis, visualisation, writing

Enno Schefuß: funding acquisition, conceptualization, investigation, review & editing

Jeroen Groeneveld: analysis, interpretation, methodology, review & editing

Charlotte Miller: analysis, interpretation, review & editing

Matthias Zabel: funding acquisition, project administration, conceptualization, investigation,

review & editing

Sample and data availability

Samples and data are respectively archived at the GeoB Core Repository and Pangaea

(www.pangaea.de) both located at MARUM, University of Bremen.

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

| core | material | depth (cm) | lab num | $^{14}$C uncalib. | cal. age yrs BP -2s | +2s | median |
|---|---|---|---|---|---|---|---|
| GeoB20616-1 | bulk | 0.5 | Poz-890 | 1640 ± 30 BP | 972 | 1168 | 1075 |
| GeoB20616-1 | *Globigerinoides ruber* | 2 | Poz-886 | 2860 ± 90 BP | 2262 | 2718 | 2473 |
| GeoB20616-1 | *Globigerinoides ruber* | 52 | Poz-890 | 5860 ± 150 BP | 5794 | 5796 | 6139 |
| GeoB20616-1 | *Globigerinoides ruber* | 102 | Poz-890 | 14290 ± 200 BP | 16063 | 17248 | 16648 |
| GeoB20616-1 | *Globigerinoides ruber* | 152 | Poz-890 | 13960 ± 390 BP | 15047 | 17400 | 16170 |
| GeoB20616-1 | *Globigerinoides ruber* | 202 | Poz-890 | 19160 ± 200 BP | 22002 | 22971 | 22511 |
| GeoB20616-1 | *Globigerinoides ruber* | 252 | Poz-889 | 20370 ± 220 BP | 23342 | 24413 | 23877 |
| GeoB20616-1 | *Globigerinoides ruber* | 302 | Poz-890 | 22070 ± 220 BP | 25365 | 26216 | 25826 |
| GeoB20616-1 | *Globigerinoides ruber* | 352 | Poz-886 | 30850 ± 870 BP | 32455 | 36152 | 34343 |
| GeoB 20616-1 | shell fragment | 390 | Poz-850 | 35820 ± 520 BP | 38724 | 41007 | 39859 |
| GeoB 20616-1 | gastropod | 634 | Poz-850 | >52000 BP | Date out of range | | |
| GeoB 20616 -1 | coral | 664 | Poz-850 | >48000 BP | Date out of range | | |

# A

SST [°C]

Limpopo
catchment

Incomati,
Matola,
Lusutfu
catchment

CC

AC

MC

## vegetation

Zambezian Mopane
savannah

Highfeld grasslands

Tongaland-Pondoland
woodland and coastal forest

Afromontane forest

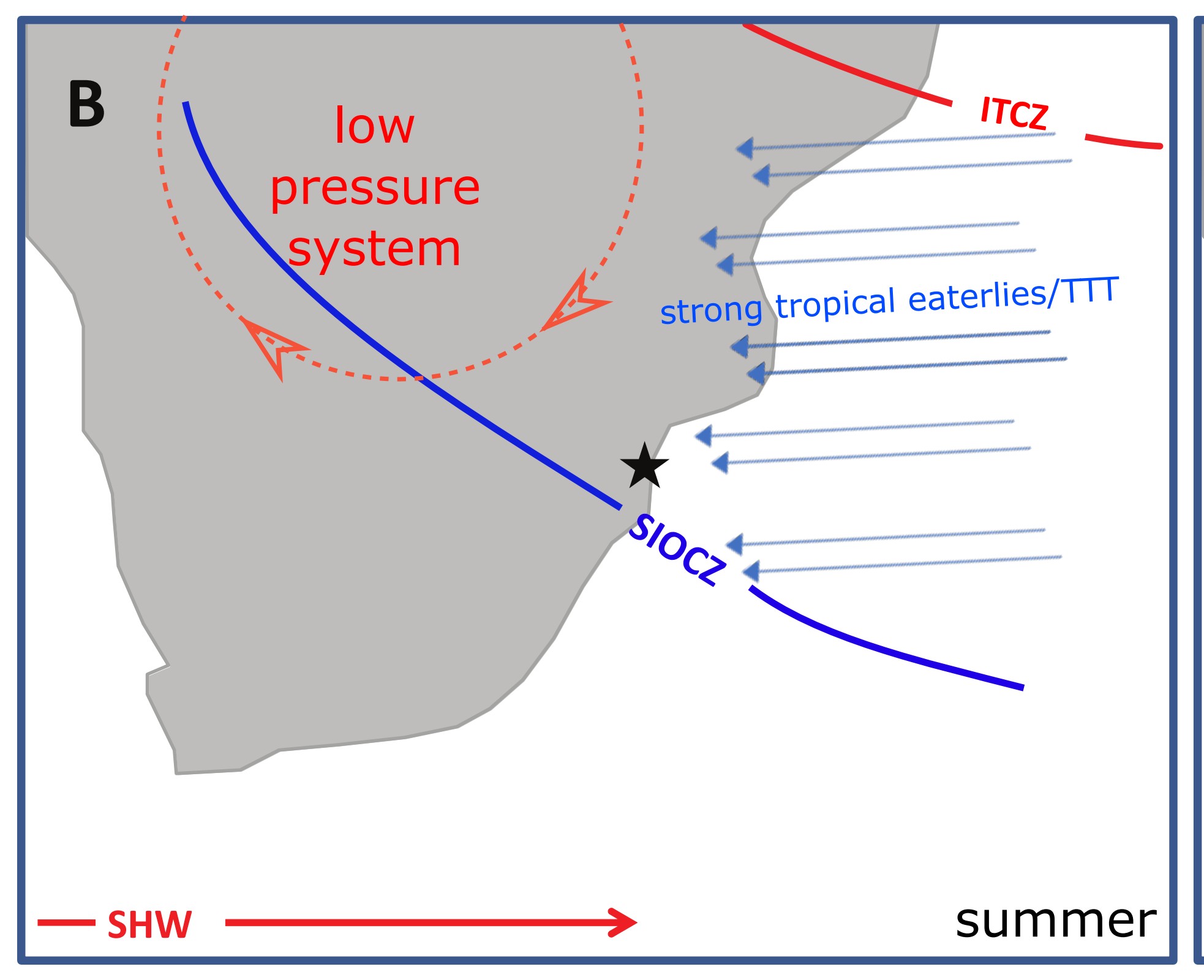

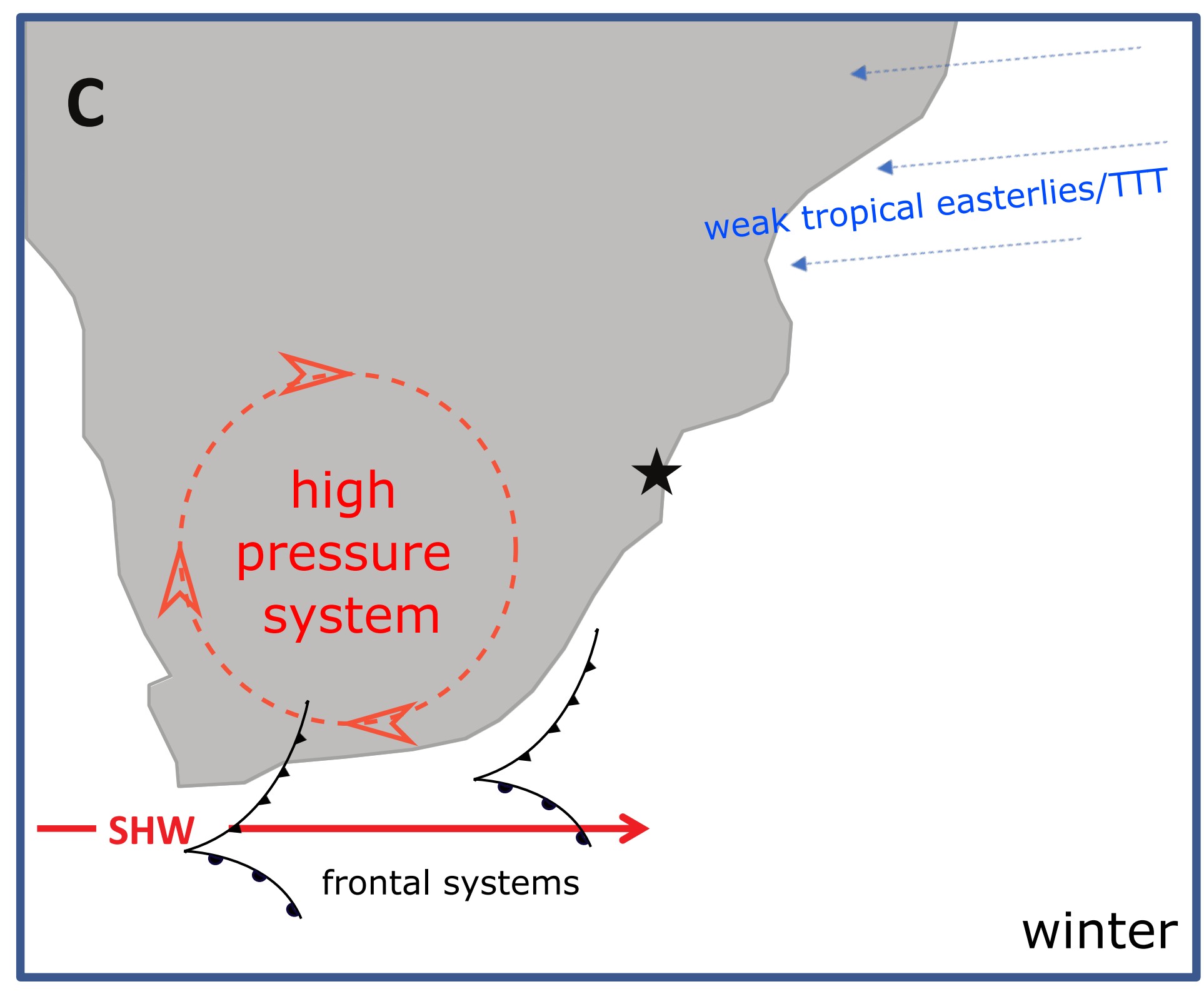

Figure 2a)

Figure 2b

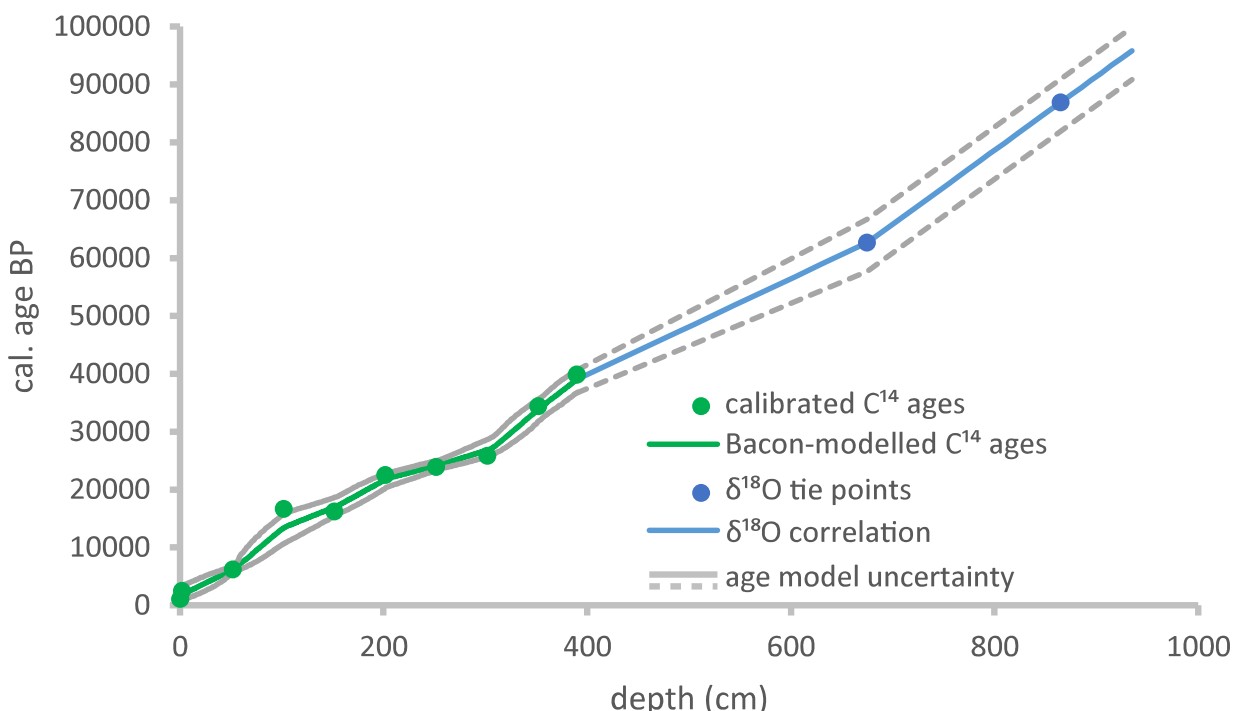

Figure 3

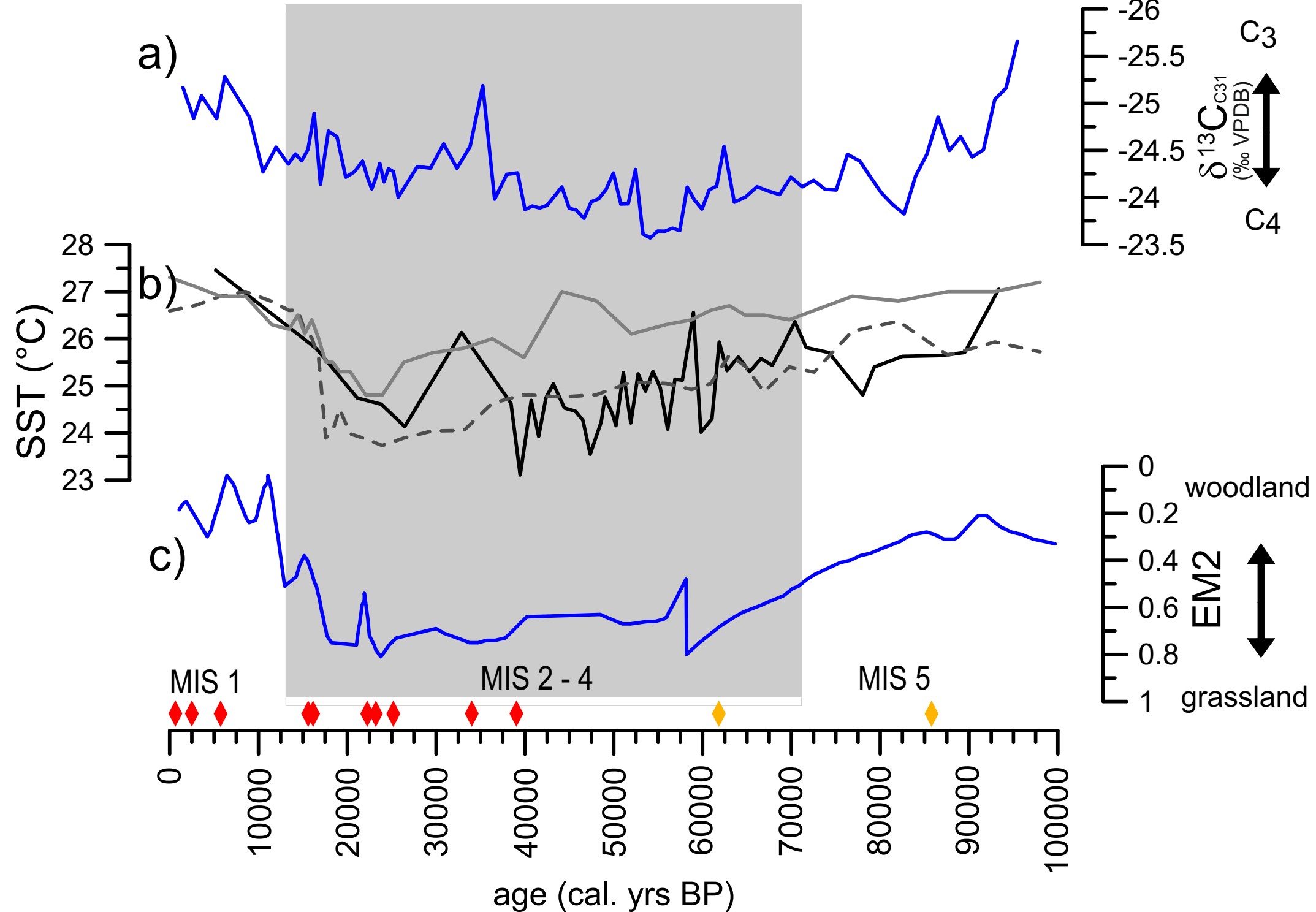

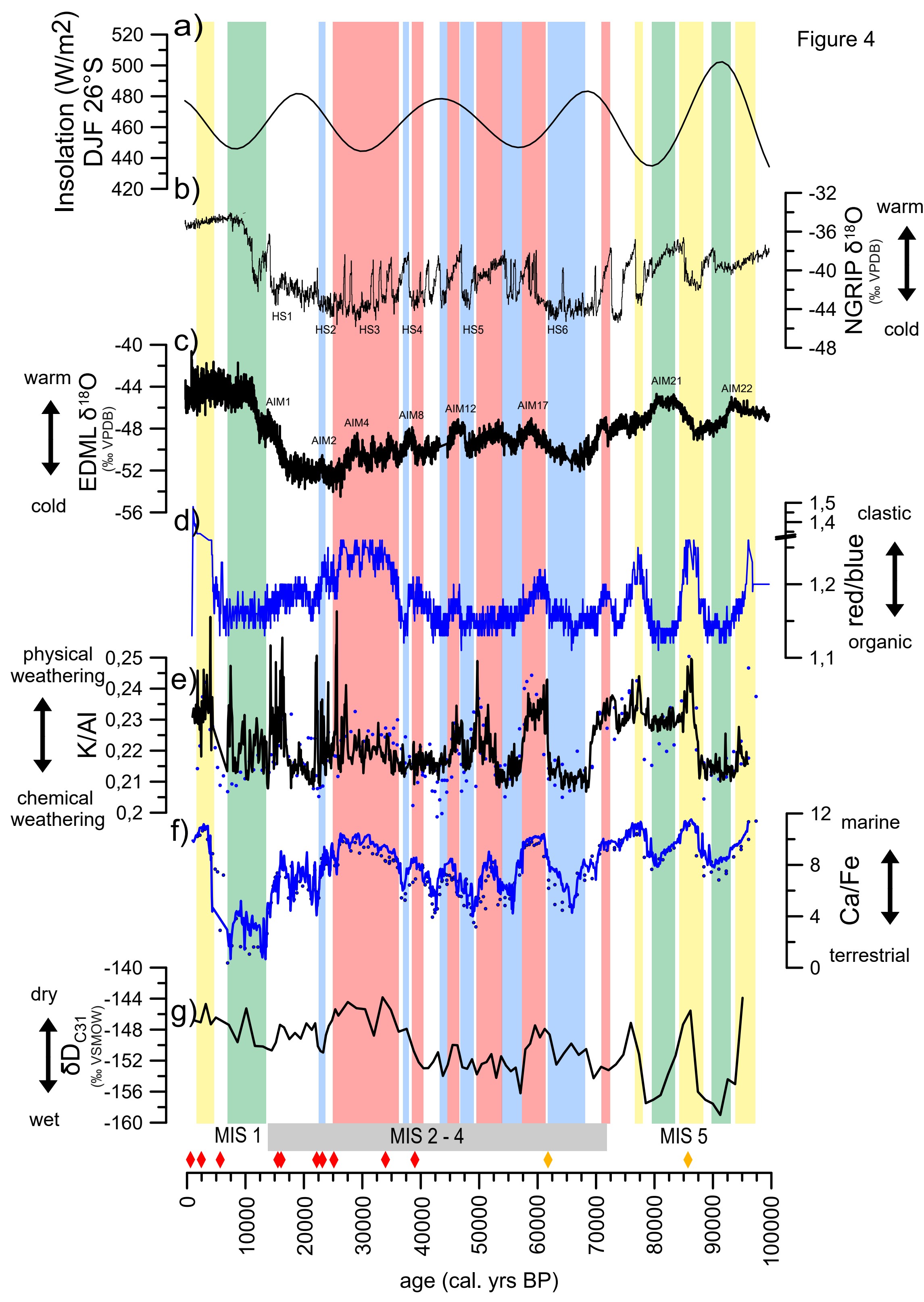

Figure 4

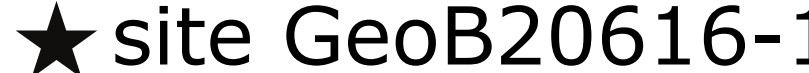

**A**

ITCZ

SR

SIOCZ

HPC

WR

precipitations shifts

during "decompressed"

**interglacial state**

SHW

**B**

*southward shifts of the thermal equator and the ITCZ*

ITCZ

SR

SIOCZ

WR

SHW

*Northward shift caused by Antartic sea-ice extent*

precipitations shifts

during "compressed"

**glacial state**

★ site GeoB20616-1

Supplement 1: GeoB20616-1 Oxygen and carbon
isotopic composition of planktonic foraminifera.

| age (cal. yrs BP) | depth (cm) | $\delta^{18}O$ (‰ VPDB) | $\delta^{13}C$ (‰ VPDB) |
|---|---|---|---|
| 37173 | 395 | -0.44 | 0.554 |
| 45285 | 455 | -0.65 | 0.709 |
| 50017 | 490 | -0.58 | 0.276 |
| 52045 | 505 | -0.72 | 0.669 |
| 57453 | 545 | -0.63 | 0.622 |
| 65565 | 605 | -0.73 | 0.435 |
| 68945 | 630 | -0.87 | 0.624 |
| 75029 | 675 | -0.35 | 0.267 |
| 77733 | 695 | -0.39 | 1.277 |
| 85845 | 755 | -1.03 | 0.954 |
| 87197 | 765 | -0.93 | 0.719 |
| 88549 | 775 | -1.03 | 0.683 |
| 89901 | 785 | -1.00 | 0.267 |
| 91253 | 795 | -0.86 | 0.772 |
| 93281 | 810 | -1.37 | 0.519 |
| 94633 | 820 | -1.00 | 0.125 |
| 95985 | 830 | -1.25 | 1.091 |
| 96525 | 834 | -1.27 | 0.519 |
| 98013 | 845 | -1.27 | 0.561 |
| 99365 | 855 | -1.10 | 0.711 |
| 100717 | 865 | -0.77 | 0.625 |
| 102069 | 875 | -1.27 | 0.663 |
| 103421 | 885 | -1.42 | 0.939 |
| 104773 | 895 | -1.16 | 0.147 |
| 106125 | 905 | -1.66 | 0.208 |
| 107477 | 915 | -1.58 | 0.927 |
| 108829 | 925 | -1.29 | 0.616 |
| 110181 | 935 | -1.56 | 0.101 |

Supplement 2:GeoB20616-1 downcore sea surface temperatures (SST) calculated following Lea et al. 2003 using the listed ICP-OES measurement results.

| age (cal. yrs BP) | depth (cm) | Mg/Ca | SST (°C) |
|---|---|---|---|
| 5179 | 40.5 | 4.38 | 27.5 |
| 16512 | 145.5 | 3.77 | 25.8 |
| 21125 | 195.5 | 3.44 | 24.7 |
| 23832 | 245.5 | 3.40 | 24.6 |
| 26461 | 295.5 | 3.26 | 24.1 |
| 32869 | 345.5 | 3.89 | 26.1 |
| 38427 | 385.5 | 3.40 | 24.6 |
| 39467 | 395.5 | 2.97 | 23.1 |
| 40712 | 410.5 | 3.42 | 24.7 |
| 41542 | 420.5 | 3.20 | 23.9 |
| 42372 | 430.5 | 3.44 | 24.7 |
| 43202 | 440.5 | 3.53 | 25.0 |
| 44447 | 455.5 | 3.37 | 24.5 |
| 45692 | 470.5 | 3.35 | 24.5 |
| 46522 | 480.5 | 3.30 | 24.3 |
| 47352 | 490.5 | 3.09 | 23.5 |
| 48597 | 505.5 | 3.28 | 24.2 |
| 49012 | 510.5 | 3.44 | 24.8 |
| 49842 | 520.5 | 3.33 | 24.4 |
| 50258 | 525.5 | 3.26 | 24.2 |
| 51088 | 535.5 | 3.60 | 25.3 |
| 51918 | 545.5 | 3.28 | 24.2 |
| 52748 | 555.5 | 3.60 | 25.3 |
| 53578 | 565.5 | 3.48 | 24.9 |
| 54408 | 575.5 | 3.61 | 25.3 |
| 55238 | 585.5 | 3.50 | 25.0 |
| 56068 | 595.5 | 3.24 | 24.1 |
| 56898 | 605.5 | 3.56 | 25.1 |
| 57728 | 615.5 | 3.55 | 25.1 |
| 58973 | 630.5 | 4.04 | 26.6 |
| 59803 | 640.5 | 3.22 | 24.0 |
| 61048 | 655.5 | 3.30 | 24.3 |
| 61878 | 665.5 | 3.82 | 25.9 |
| 62748 | 675.5 | 3.62 | 25.3 |
| 64023 | 685.5 | 3.71 | 25.6 |
| 65298 | 695.5 | 3.61 | 25.3 |
| 66573 | 705.5 | 3.70 | 25.6 |
| 67848 | 715.5 | 3.66 | 25.4 |
| 69123 | 725.5 | 3.80 | 25.9 |
| 70397 | 735.5 | 3.97 | 26.4 |
| 71672 | 745.5 | 3.78 | 25.8 |
| 74222 | 765.5 | 3.74 | 25.7 |
| 78046 | 795.5 | 3.46 | 24.8 |
| 79321 | 805.5 | 3.64 | 25.4 |
| 82508 | 830.5 | 3.72 | 25.6 |
| 86970 | 865.5 | 3.72 | 25.6 |
| 89520 | 885.5 | 3.75 | 25.7 |
| 93344 | 915.5 | 4.22 | 27.1 |

Supplement 3: GeoB20616-1 organic geochemical down-core data. n-Alkane isotopic composition and distribution descriptive parameters averaged. The elevated CPI values ranging from 3.8 to 14 indicate that the *n* alkanes within the terrestrial and marine samples were likely derived from nondegraded, terrestrial, higher plant material (Eglinton & Hamilton, 1967). We focus the discussion on the isotopic signals of the *n* -C31 alkane but note that the *n* -C29 and *n* -C33 alkanes reveal similar trends.

| age (cal. yrs BP) | Depth (cm) | $\delta^{13}$C-$n$C$_{29}$ (‰ VPDB) | $\delta^{13}$C-$n$C$_{31}$ (‰ VPDB) | $\delta^{13}$C-$n$C$_{33}$ (‰ VPDB) | $\delta$D-$n$C$_{29}$ (‰ VSMOW) | $\delta$D-$n$C$_{31}$ (‰ VSMOW) | $\delta$D-$n$C$_{33}$ (‰ VSMOW) | CPI$_{25-33}$ |
|---|---|---|---|---|---|---|---|---|
| 1504 | 2 | -26.0 | -25.2 | -23.3 | -133 | -147 | -150 | 6 |
| 2705 | 12 | -25.6 | -24.8 | -23.2 | -140 | -147 | -150 | 6 |
| 3580 | 22 | -26.2 | -25.1 | -23.3 | -138 | -145 | -148 | 7 |
| 4442 | 32 | -26.1 | -25.0 | -23.2 | -144 | -147 | -151 | 7 |
| 5306 | 42 | -26.0 | -24.8 | -23.0 | -130 | -146 | -148 | 7 |
| 6197 | 52 | -25.7 | -25.3 | -23.2 | | | | 7 |
| 7535 | 62 | | | | -140 | -147 | -148 | 7 |
| 9007 | 72 | -25.9 | -24.9 | -23.0 | -137 | -150 | -150 | 8 |
| 10513 | 82 | -25.2 | -24.3 | -22.5 | -134 | -145 | -149 | 7 |
| 11985 | 92 | -24.9 | -24.5 | -22.4 | -136 | -150 | -151 | 8 |
| 13363 | 102 | -25.1 | -24.4 | -22.5 | -141 | -150 | -151 | 7 |
| 14163 | 112 | -25.2 | -24.5 | -22.6 | -136 | -150 | -150 | 7 |
| 14871 | 122 | -25.2 | -24.4 | -22.5 | -145 | -151 | -151 | 8 |
| 15571 | 132 | -25.4 | -24.5 | -22.7 | -135 | -150 | -149 | 7 |
| 16271 | 142 | -25.5 | -24.9 | -22.8 | -141 | -147 | -147 | 7 |
| 16979 | 152 | -24.8 | -24.1 | -22.2 | -139 | -148 | -145 | 7 |
| 17878 | 162 | -25.0 | -24.7 | -22.6 | -143 | -149 | -149 | 8 |
| 18839 | 172 | -25.2 | -24.6 | -22.6 | -135 | -148 | -149 | 7 |
| 19822 | 182 | -24.9 | -24.2 | -22.3 | -138 | -149 | -151 | 7 |
| 20787 | 192 | -25.0 | -24.3 | -22.4 | -139 | -147 | -147 | 7 |
| 21724 | 202 | -25.1 | -24.4 | -22.5 | -138 | -148 | -148 | 7 |
| 22300 | 212 | -25.0 | -24.2 | -22.4 | -130 | -147 | -148 | 7 |
| 22756 | 222 | -24.7 | -24.1 | -22.2 | -137 | -150 | -150 | 11 |
| 23217 | 232 | -25.0 | -24.2 | -22.4 | -135 | -151 | -149 | 8 |
| 23673 | 242 | -25.5 | -24.4 | -22.5 | -135 | -151 | -150 | 7 |
| 24131 | 252 | -24.7 | -24.2 | -22.3 | -132 | -149 | -148 | 7 |
| 24642 | 262 | -25.3 | -24.3 | -22.5 | -132 | -148 | -148 | 7 |
| 25187 | 272 | -25.4 | -24.3 | -22.4 | -131 | -147 | -148 | 7 |
| 25731 | 282 | -25.0 | -24.0 | -22.3 | -135 | -145 | -149 | 7 |
| 26273 | 292 | | | | -138 | -146 | -148 | 8 |
| 27880 | 312 | -25.2 | -24.3 | -22.4 | -138 | -144 | -148 | 7 |
| 29364 | 322 | -25.4 | -24.3 | -22.6 | -130 | -145 | -147 | 7 |
| 30853 | 332 | -25.4 | -24.6 | -22.6 | -132 | -145 | -147 | 7 |
| 32351 | 342 | -25.3 | -24.3 | -22.5 | -142 | -149 | -151 | 7 |
| 33833 | 352 | -25.5 | -24.5 | -22.6 | -127 | -144 | -147 | 7 |
| 35256 | 362 | -25.9 | -25.2 | -23.2 | -141 | -145 | -150 | 8 |
| 36591 | 372 | -25.1 | -24.0 | -22.1 | -133 | -148 | -152 | 8 |
| 37949 | 382 | -25.1 | -24.2 | -22.3 | -132 | -148 | -149 | 7 |
| 39177 | 392 | -25.0 | -24.3 | -22.6 | -143 | -151 | -153 | 7 |
| 40007 | 402 | -24.9 | -23.9 | -22.2 | -137 | -152 | -152 | 8 |
| 40837 | 412 | -24.8 | -23.9 | -22.1 | -143 | -153 | -155 | 7 |
| 41667 | 422 | -24.8 | -23.9 | -22.0 | -144 | -153 | -153 | 8 |
| 42497 | 432 | -24.9 | -23.9 | -22.1 | -142 | -152 | -154 | 8 |
| 43327 | 442 | -25.0 | -24.0 | -22.2 | -136 | -151 | -152 | 7 |
| 44157 | 452 | -25.3 | -24.1 | -22.4 | -138 | -154 | -154 | 7 |
| 44987 | 462 | -24.9 | -23.9 | -22.2 | -137 | -153 | -154 | 7 |
| 45817 | 472 | -25.0 | -23.9 | -22.0 | -143 | -150 | -152 | 7 |
| 46647 | 482 | -24.7 | -23.8 | -22.0 | -145 | -150 | -152 | 6 |
| 47477 | 492 | -25.0 | -24.0 | -22.1 | -138 | -153 | -154 | 7 |
| 48307 | 502 | -24.7 | -24.0 | -22.1 | -142 | -153 | -153 | 7 |
| 49137 | 512 | -25.0 | -24.1 | -22.1 | -145 | -152 | -153 | 7 |
| 49967 | 522 | -24.9 | -24.3 | -22.2 | -143 | -154 | -155 | 7 |
| 50797 | 532 | -24.9 | -23.9 | -22.0 | -142 | -152 | -154 | 7 |
| 51627 | 542 | -24.9 | -23.9 | -22.1 | -141 | -152 | -155 | 6 |
| 52457 | 552 | -24.1 | -24.3 | -22.7 | -139 | -151 | -154 | 7 |
| 53287 | 562 | -24.7 | -23.6 | -22.0 | -141 | -154 | -155 | 9 |
| 54117 | 572 | -24.4 | -23.6 | -21.9 | -137 | -151 | -153 | 7 |
| 54947 | 582 | -24.6 | -23.6 | -22.1 | -147 | -153 | -154 | 7 |
| 55777 | 592 | -24.3 | -23.6 | -22.0 | -147 | -153 | -155 | 7 |
| 56607 | 602 | -24.8 | -23.7 | -21.9 | -138 | -153 | -155 | 6 |
| 57437 | 612 | -24.7 | -23.6 | -22.1 | -147 | -156 | -156 | 7 |
| 58267 | 622 | -25.1 | -24.1 | -22.3 | -145 | -151 | -152 | 7 |
| 59097 | 632 | -25.1 | -24.0 | -22.3 | -138 | -150 | -152 | 7 |
| 59927 | 642 | -25.0 | -23.9 | -22.1 | -134 | -147 | -148 | 6 |
| 60757 | 652 | -25.1 | -24.1 | -22.4 | -139 | -149 | -148 | 7 |
| 61588 | 662 | -25.1 | -24.1 | -22.4 | -136 | -148 | -149 | 7 |
| 62418 | 672 | -25.4 | -24.5 | -22.8 | -132 | -149 | -148 | 7 |
| 63577 | 682 | -25.4 | -23.9 | -22.5 | -146 | -153 | -153 | 8 |
| 64852 | 692 | -24.8 | -24.0 | -22.4 | -134 | -151 | -152 | 7 |
| 66127 | 702 | -25.0 | -24.1 | -22.6 | -134 | -150 | -151 | 7 |
| 67402 | 712 | -24.8 | -24.1 | -22.5 | -134 | -151 | -153 | 7 |
| 68676 | 722 | -25.1 | -24.0 | -22.6 | -144 | -151 | -152 | 7 |
| 69951 | 732 | -25.3 | -24.2 | -22.8 | -138 | -154 | -156 | 7 |
| 71226 | 742 | -25.6 | -24.1 | -22.7 | -138 | -153 | -155 | 7 |
| 72501 | 752 | -25.4 | -24.2 | -22.6 | -138 | -153 | -156 | 7 |
| 73776 | 762 | -25.4 | -24.1 | -22.6 | -140 | -152 | -153 | 7 |
| 75050 | 772 | -25.4 | -24.1 | -22.6 | -145 | -151 | -153 | 7 |
| 76325 | 782 | -26.3 | -24.5 | -23.0 | -137 | -147 | -150 | 6 |
| 77728 | 793 | -25.8 | -24.4 | -22.7 | -142 | -151 | -154 | 8 |
| 78875 | 802 | -25.2 | -24.2 | -22.3 | -153 | -157 | -159 | 7 |
| 80150 | 812 | -25.3 | -24.0 | -22.3 | -137 | -157 | -159 | 6 |
| 81425 | 822 | -25.3 | -23.9 | -22.3 | -149 | -156 | -159 | 6 |
| 82699 | 832 | -24.0 | -23.8 | -22.2 | -137 | -154 | -155 | 5 |
| 83974 | 842 | -25.5 | -24.2 | -22.6 | -144 | -151 | -153 | 6 |
| 85249 | 852 | -25.6 | -24.5 | -22.8 | -141 | -147 | -152 | 7 |
| 86524 | 862 | -26.2 | -24.9 | -23.1 | -142 | -146 | -151 | 1 |
| 87799 | 872 | -25.9 | -24.5 | -22.8 | -146 | -156 | -157 | 6 |
| 89074 | 882 | -26.2 | -24.6 | -22.9 | -147 | -157 | -157 | 6 |
| 90348 | 892 | -25.8 | -24.4 | -22.7 | -151 | -158 | -161 | 7 |
| 91623 | 902 | -26.1 | -24.5 | -22.9 | -150 | -159 | -160 | 6 |
| 92898 | 912 | -26.4 | -25.0 | -22.9 | -139 | -154 | -157 | 7 |
| 94173 | 922 | -26.6 | -25.2 | -23.1 | -141 | -155 | -158 | 7 |
| 95448 | 932 | -26.3 | -25.7 | -23.6 | -134 | -144 | -146 | 7 |

Supplement 4: GeoB20616-1 inorganic geochemical down-core data from

| depth (cm) | age (cal. years BP) | Al (mg/kg) | Ca (mg/kg) | Fe (mg/kg) | K (mg/kg) |
|---|---|---|---|---|---|
| 5.5 | 2141 | 41601 | 177248 | 27139 | 8911 |
| 10.5 | 2575 | 40009 | 184378 | 26882 | 8863 |
| 15.5 | 3008 | 33894 | 202638 | 23765 | 8045 |
| 20.5 | 3446 | 32489 | 209794 | 22430 | 7864 |
| 25.5 | 3891 | 42712 | 174836 | 30180 | 10010 |
| 30.5 | 4315 | 51353 | 145437 | 36638 | 11646 |
| 35.5 | 4741 | 52964 | 143556 | 38025 | 12066 |
| 40.5 | 5179 | 61319 | 114217 | 41607 | 13549 |
| 45.5 | 5600 | 62755 | 114701 | 43780 | 13466 |
| 50.5 | 6032 | 76367 | 67954 | 51567 | 15949 |
| 55.5 | 6602 | 64118 | 106827 | 43037 | 13586 |
| 60.5 | 7317 | 81808 | 55271 | 53594 | 16917 |
| 65.5 | 8040 | 78369 | 59242 | 50854 | 16399 |
| 70.5 | 8781 | 75738 | 70803 | 48374 | 15809 |
| 75.5 | 9523 | 78203 | 60655 | 52427 | 16661 |
| 80.5 | 10290 | 78352 | 59011 | 51635 | 16528 |
| 85.5 | 11035 | 77321 | 59370 | 54169 | 16743 |
| 90.5 | 11767 | 76465 | 63916 | 55399 | 16351 |
| 95.5 | 12501 | 75170 | 61400 | 53243 | 16427 |
| 100.5 | 13211 | 77240 | 61674 | 55229 | 16505 |
| 105.5 | 13696 | 74527 | 68091 | 55522 | 16528 |
| 110.5 | 14057 | 74176 | 67078 | 52330 | 15914 |
| 115.5 | 14409 | 71905 | 75806 | 48928 | 15316 |
| 120.5 | 14766 | 68726 | 82184 | 47674 | 15031 |
| 125.5 | 15118 | 64537 | 95917 | 45311 | 13962 |
| 130.5 | 15468 | 61688 | 99668 | 41571 | 13308 |
| 135.5 | 15813 | 59928 | 106381 | 41391 | 13475 |
| 140.5 | 16168 | 62339 | 102200 | 42106 | 13815 |
| 145.5 | 16512 | 53917 | 124124 | 38125 | 12401 |
| 150.5 | 16868 | 53583 | 129426 | 38349 | 12385 |
| 155.5 | 17255 | 62403 | 104697 | 43747 | 13950 |
| 160.5 | 17734 | 66270 | 88069 | 48473 | 14453 |
| 165.5 | 18215 | 56286 | 115156 | 39825 | 13046 |
| 170.5 | 18696 | 67792 | 80244 | 44942 | 14654 |
| 175.5 | 19176 | 66981 | 88478 | 45807 | 14197 |
| 180.5 | 19674 | 62491 | 95931 | 42658 | 13873 |
| 185.5 | 20162 | 65334 | 99870 | 47296 | 14113 |
| 190.5 | 20643 | 62511 | 102887 | 41707 | 13467 |
| 195.5 | 21125 | 66868 | 96885 | 48235 | 14988 |
| 200.5 | 21597 | 67068 | 89420 | 49711 | 15021 |
| 205.5 | 22000 | 69028 | 90573 | 44979 | 14341 |
| 210.5 | 22233 | 60283 | 114739 | 42526 | 13499 |
| 215.5 | 22458 | 69104 | 88862 | 45349 | 14645 |
| 220.5 | 22687 | 70456 | 88409 | 45770 | 14638 |
| 225.5 | 22917 | 72078 | 84770 | 46615 | 14792 |
| 230.5 | 23146 | 65131 | 103541 | 43173 | 13557 |
| 235.5 | 23379 | 56941 | 129414 | 38257 | 12407 |
| 240.5 | 23605 | 60846 | 120300 | 40670 | 12692 |
| 245.5 | 23832 | 57609 | 129981 | 41183 | 12521 |
| 250.5 | 24063 | 58536 | 123484 | 41087 | 12795 |
| 255.5 | 24298 | 58042 | 123364 | 40032 | 12580 |
| 260.5 | 24562 | 57797 | 129772 | 43387 | 13060 |
| 280.5 | 25648 | 60102 | 118404 | 40438 | 13120 |
| 285.5 | 25922 | 59111 | 122979 | 41101 | 12896 |
| 290.5 | 26193 | 60619 | 123278 | 41523 | 12833 |
| 295.5 | 26461 | 48348 | 167591 | 34485 | 10447 |
| 300.5 | 26733 | 44819 | 174138 | 31605 | 10008 |
| 305.5 | 27045 | 44879 | 172727 | 32203 | 10102 |
| 310.5 | 27652 | 45798 | 169598 | 32696 | 10287 |
| 315.5 | 28407 | 51376 | 146621 | 36184 | 11618 |
| 320.5 | 29142 | 45342 | 164907 | 31759 | 10547 |
| 325.5 | 29884 | 54584 | 140547 | 36636 | 12082 |
| 330.5 | 30625 | 49338 | 155126 | 34580 | 11003 |
| 335.5 | 31378 | 49564 | 154649 | 34299 | 11218 |
| 340.5 | 32130 | 52464 | 140437 | 37184 | 11815 |
| 345.5 | 32869 | 53070 | 137093 | 36736 | 11702 |
| 350.5 | 33615 | 53940 | 137642 | 38276 | 12143 |
| 355 | 34273 | 55917 | 113821 | 35189 | 12547 |
| 355.5 | 34346 | 56963 | 129853 | 40731 | 12948 |
| 360 | 34982 | 56742 | 111906 | 36524 | 12790 |
| 360 | 34982 | 51631 | 124407 | 35717 | 11300 |
| 360.5 | 35050 | 57729 | 127739 | 40217 | 12990 |
| 365 | 35663 | 55346 | 112114 | 34237 | 12311 |
| 365.5 | 35731 | 56217 | 128293 | 39866 | 12775 |
| 370 | 36319 | 52984 | 117699 | 35315 | 11500 |
| 370 | 36319 | 54704 | 116866 | 36474 | 12137 |
| 370.5 | 36386 | 57846 | 110492 | 39138 | 13301 |
| 375.5 | 37069 | 66718 | 83800 | 46061 | 14718 |
| 380 | 37677 | 66930 | 72739 | 41065 | 14458 |
| 380 | 37677 | 64675 | 79088 | 44574 | 13757 |
| 380.5 | 37744 | 65197 | 85613 | 44766 | 14634 |
| 385.5 | 38427 | 64755 | 94671 | 44168 | 14123 |
| 390 | 39011 | 63041 | 93578 | 42042 | 12755 |
| 390.5 | 39052 | 64839 | 97063 | 45540 | 13991 |
| 395.5 | 39467 | 64293 | 101333 | 43352 | 13659 |
| 400 | 39841 | 56336 | 111917 | 37771 | 11815 |
| 400.5 | 39882 | 59540 | 116418 | 40174 | 12958 |
| 405 | 40256 | 60525 | 101730 | 35780 | 13075 |
| 405.5 | 40297 | 60053 | 113436 | 41620 | 13286 |
| 410 | 40671 | 56134 | 111014 | 37702 | 11909 |
| 410.5 | 40712 | 61267 | 115181 | 40783 | 13059 |
| 415.5 | 41127 | 68070 | 96946 | 45136 | 14410 |
| 420 | 41501 | 60132 | 92248 | 41222 | 12966 |
| 420.5 | 41542 | 63673 | 107364 | 44591 | 13908 |
| 425 | 41916 | 64280 | 79834 | 42446 | 14274 |
| 425.5 | 41957 | 67059 | 90673 | 43957 | 14495 |
| 430 | 42331 | 66538 | 74768 | 44034 | 13772 |
| 430.5 | 42372 | 74425 | 72721 | 49023 | 16000 |
| 435 | 42746 | 68276 | 73664 | 46167 | 14581 |
| 435.5 | 42787 | 74586 | 74830 | 48272 | 15428 |
| 440 | 43161 | 68894 | 68864 | 42636 | 13686 |
| 440.5 | 43202 | 73481 | 85417 | 52971 | 15420 |
| 445 | 43576 | 63726 | 90121 | 43258 | 13714 |
| 445.5 | 43617 | 70805 | 90745 | 47880 | 14455 |
| 450 | 43991 | 62794 | 94709 | 41966 | 12541 |

| | | | | | |
|---|---|---|---|---|---|
| 450.5 | 44032 | 66645 | 102400 | 44605 | 14024 |
| 455.5 | 44447 | 63058 | 110000 | 43305 | 13270 |
| 460 | 44821 | 60234 | 103253 | 41532 | 12445 |
| 460 | 44821 | 63467 | 94432 | 41145 | 13774 |
| 460.5 | 44862 | 66633 | 99157 | 45814 | 14019 |
| 465.5 | 45277 | 63091 | 106734 | 43179 | 13627 |
| 470 | 45651 | 58903 | 94189 | 39768 | 12457 |
| 470.5 | 45692 | 63933 | 103868 | 45098 | 14305 |
| 475 | 46066 | 60994 | 86668 | 37042 | 13787 |
| 475.5 | 46107 | 61837 | 98873 | 41100 | 14153 |
| 480 | 46481 | 56561 | 90832 | 40272 | 12613 |
| 480.5 | 46522 | 66192 | 89069 | 45694 | 14571 |
| 485.5 | 46937 | 67609 | 86077 | 46304 | 14804 |
| 490 | 47311 | 62425 | 78589 | 42526 | 13456 |
| 490.5 | 47352 | 73996 | 74298 | 50343 | 15312 |
| 495 | 47726 | 63854 | 82115 | 42127 | 14185 |
| 495.5 | 47767 | 70040 | 79558 | 47674 | 14918 |
| 500 | 48141 | 58443 | 92436 | 39597 | 12792 |
| 500.5 | 48182 | 67806 | 89654 | 41300 | 14255 |
| 505.5 | 48597 | 64302 | 95964 | 42143 | 13817 |
| 510 | 48971 | 56298 | 89156 | 36482 | 12774 |
| 510.5 | 49012 | 58933 | 107154 | 37850 | 13198 |
| 515.5 | 49427 | 56637 | 121832 | 39000 | 13166 |
| 520 | 49801 | 69152 | 66613 | 44369 | 14418 |
| 520.5 | 49842 | 61249 | 102619 | 40085 | 14223 |
| 525.5 | 50258 | 59082 | 98837 | 38056 | 13993 |
| 530 | 50631 | 63717 | 71262 | 40987 | 13969 |
| 530.5 | 50673 | 62272 | 86967 | 41175 | 14570 |
| 535.5 | 51088 | 66636 | 77489 | 41972 | 15207 |
| 540 | 51461 | 67956 | 65810 | 40760 | 14770 |
| 540.5 | 51503 | 68530 | 79364 | 46688 | 15272 |
| 545.5 | 51918 | 71167 | 71372 | 47010 | 16052 |
| 550.5 | 52333 | 69486 | 77430 | 47457 | 15274 |
| 555.5 | 52748 | 73000 | 67616 | 47939 | 15957 |
| 560 | 53121 | 70124 | 57144 | 42084 | 15288 |
| 560 | 53121 | 65831 | 79730 | 43561 | 13966 |
| 560.5 | 53163 | 70140 | 86254 | 47706 | 15075 |
| 565.5 | 53578 | 72087 | 78510 | 46632 | 15068 |
| 570 | 53951 | 66312 | 82360 | 42526 | 14264 |
| 570 | 53951 | 66194 | 76845 | 43042 | 13921 |
| 580 | 54781 | 66247 | 80417 | 44078 | 14268 |
| 580 | 54781 | 68014 | 77544 | 46462 | 13950 |
| 590 | 55611 | 66355 | 83582 | 44236 | 14347 |
| 590 | 55611 | 65058 | 82253 | 45714 | 13717 |
| 600 | 56441 | 61234 | 89130 | 40404 | 13039 |
| 610 | 57271 | 55032 | 110938 | 38905 | 12495 |
| 610 | 57271 | 58564 | 100873 | 40606 | 12782 |
| 620 | 58101 | 45455 | 139095 | 31370 | 10910 |
| 620 | 58101 | 44378 | 145060 | 31074 | 10138 |
| 625 | 58516 | 43060 | 142267 | 29805 | 10433 |
| 630 | 58931 | 42528 | 144291 | 29578 | 9941 |
| 635 | 59346 | 42342 | 162679 | 30808 | 10345 |
| 640 | 59761 | 43435 | 145528 | 30639 | 10129 |
| 645 | 60176 | 43778 | 146212 | 29584 | 10425 |
| 650 | 60591 | 42732 | 148481 | 29140 | 10062 |
| 660 | 61422 | 51553 | 125405 | 35513 | 11277 |
| 670 | 62252 | 61084 | 96695 | 42103 | 12882 |
| 670 | 62252 | 64483 | 93359 | 45537 | 13816 |
| 680 | 63322 | 67129 | 77902 | 46161 | 14021 |
| 680 | 63322 | 62970 | 93170 | 43272 | 13659 |
| 690 | 64597 | 67932 | 76483 | 44205 | 13923 |
| 690 | 64597 | 68020 | 73796 | 46466 | 14663 |
| 700 | 65872 | 67826 | 80960 | 48905 | 14043 |
| 710 | 67147 | 58768 | 103678 | 40525 | 12421 |
| 710 | 67147 | 62192 | 95677 | 50634 | 13671 |
| 720 | 68421 | 62300 | 100698 | 43941 | 13308 |
| 720 | 68421 | 59517 | 102095 | 41114 | 12571 |
| 730 | 69696 | 51990 | 130019 | 36401 | 11173 |
| 730 | 69696 | 60917 | 103143 | 43866 | 13069 |
| 740 | 70971 | 42149 | 152024 | 32335 | 9772 |
| 740 | 70971 | 48978 | 143668 | 38215 | 11380 |
| 750 | 72246 | 48501 | 141805 | 34406 | 11020 |
| 760 | 73521 | 45861 | 150455 | 33980 | 10725 |
| 760 | 73521 | 47125 | 143183 | 34141 | 10539 |
| 770 | 74795 | 42528 | 162159 | 31551 | 10067 |
| 770 | 74795 | 39344 | 169696 | 30045 | 9127 |
| 780 | 76070 | 34419 | 192887 | 25895 | 8236 |
| 780 | 76070 | 31427 | 202857 | 23010 | 7502 |
| 790 | 77345 | 35777 | 183667 | 26151 | 8677 |
| 790 | 77345 | 35244 | 183499 | 24784 | 8237 |
| 800 | 78620 | 51577 | 132777 | 38252 | 11201 |
| 810 | 79895 | 55158 | 111085 | 40599 | 11873 |
| 820 | 81170 | 50305 | 125661 | 37449 | 11178 |
| 825 | 81807 | 52630 | 126976 | 38524 | 12155 |
| 830 | 82444 | 48311 | 141565 | 35482 | 10618 |
| 835 | 83082 | 48136 | 145329 | 35219 | 11257 |
| 840 | 83719 | 47482 | 150996 | 34356 | 10533 |
| 850 | 84994 | 32646 | 192928 | 21969 | 7706 |
| 860 | 86269 | 22912 | 239594 | 17648 | 5737 |
| 860 | 86269 | 26057 | 230951 | 18877 | 6348 |
| 870 | 87544 | 28890 | 213965 | 20328 | 6753 |
| 870 | 87544 | 52404 | 134379 | 37382 | 11542 |
| 880 | 88819 | 62396 | 105431 | 42914 | 13482 |
| 880 | 88819 | 57738 | 112182 | 39138 | 11802 |
| 890 | 90093 | 54070 | 123836 | 39282 | 11467 |
| 890 | 90093 | 61059 | 108961 | 41761 | 13009 |
| 900 | 91368 | 63645 | 102706 | 44565 | 13416 |
| 900 | 91368 | 62597 | 112688 | 41009 | 13450 |
| 910 | 92643 | 61222 | 107700 | 40574 | 12949 |
| 910 | 92643 | 62867 | 108253 | 42496 | 13677 |
| 920 | 93918 | 55146 | 133032 | 37836 | 11767 |
| 920 | 93918 | 55676 | 134602 | 38464 | 12232 |
| 930 | 95193 | 51392 | 139805 | 33200 | 11087 |
| 930 | 95193 | 50522 | 153534 | 33390 | 11269 |
| 940 | 96468 | 44837 | 163431 | 28193 | 9756 |
| 940 | 96468 | 22365 | 244843 | 14395 | 5517 |
| 950 | 97742 | 21712 | 244971 | 13257 | 5156 |