# Peer review of "Glacial to interglacial climate variability in the southeastern African subtropics (25- 20°S)"

_Climate of the Past, 2019_

## Referee Comment (RC1) · Stephan Woodborne (Referee) · 7 Jun 2020

Summary

In this article, Hahn et al. report on the analysis of a 958 cm sediment core that was taken in the Delagoa Bight off southeastern Africa. The source of the sediment is argued to be from three nearby river catchments that are relatively small, and as a result of this the environmental information derived from the sediments represents a fairly clean signal (in contrast to nearby cores that sample the Limpopo River catchment which is vast and probably includes multiple climate sensitivities).

The chronology for the core is generated using 12 radiocarbon dates, of which two are beyond the limit of the method, and $\delta18O$ values from benthic foraminifera that are

compared to the LR04 benthic stack. The radiocarbon dates from the upper part of the record overlap with the $\delta$18O values from the lower part of the stack so that one of the tie $\delta$18O tie points has an apriori age assignment. The results of the age model demonstrate a relatively constant deposition over the last 100 000 years.

Ca/Mg ratios on foraminifera are used to reconstruct past sea surface temperature (SST), and this record demonstrates that SST was almost 4°C warmer during interglacials (MIS 5 & 1) that it was during glacials (MIS4-2).

The main substance of the article is the presentation of multiple geochemical tracers of terrestrial climate and the associated vegetation responses in the catchment. These include $\delta$13C C31 (terrestrial plants community indicator), red/blue ratios (organic vs. classic indicators), K/Al (chemical vs. physical weathering), Ca/Fe (terrestrial vs. marine source indicator) and $\delta$DC31 values (rainfall amount indicators). The authors identify a coherent pattern in which all of the geochemical tracers vary in concert with one another, and this is coherently argued to reflect hydrological changes in the associated river catchments.

The underlying cause for the alternation between mesic and xeric conditions in the catchment are explored through northern hemisphere forcing in the form of Heinrich Stadials, and southern hemisphere forcing in the form of Antarctic ice advances. What emerges is that the forcing during glacials and interglacials differ from one another, and this must be reconciled through synoptic scale changes in the drivers of continental rainfall (rather than insolation variability). The model that is proposed centers on the way in which the two main moisture-bearing systems, the inter-tropical convergence zone (ITCZ) and the southern hemisphere westerlies (SHW), interact and influence the development of the South African high system that is dominant influence on modern rainfall. In essence the argument is that during glacial conditions the SHW migrate northwards because of Antarctic expansion, while the thermal equator (ITCZ) migrates southward because of Arctic expansion, and the South African high system occludes. The extinction of the South African high pressure system during glacials

prevents the development of the southern Indian Ocean convergence zone (SIOCZ) and the associated temperate tropical troughs (TTT) that dominate modern summer rainfall (but please note the comment on this subject below). As a result the catchment maintained a relatively constant water balance between glacials and interglacials as the glacial loss of the TTT/SIOCZ was compensated by direct summer rainfall from the ITCZ and/or winter rainfall from the southern hemisphere westerlies (depending on the dominant Arctic vs. Antarctic forcing at the time).

Scientific merit

Notwithstanding the critique that is presented below, this manuscript makes a valuable contribution to climate science in southern Africa. The dynamics of the climate system are relevant to both future projects of climate change, and the interpretation of the rich archaeological heritage of the region. Several archaological sequences of a similar age are in close enough proximity for the climate model to be relevant. Sibudu cave and Border Cave contain evidence of mesic/xeric cycles, and they are also well dated, so there is potential to refine the glacial/interglacial climate model. Key palaeoclimate records, such as the Pretoria Salt Pan have been dated using insolation arguments, and if the climate model proposed by Hahn et al. is correct, then the basis for the age model of the Pretoria Salt Pan is flawed.

The authors have presented their data in supplementary tables, which is going to be very useful for comparing this dataset to others.

Review

The critique of the manuscript is in the form of substantive clarifications, minor issues, and typos.

Substantive clarification

One of the most important contributions of this manuscript is the model for synoptic shifts in the region during glacial periods, and in particular its effect on the SIOCZ.

Clarification is required of exactly what the SIOCZ is. Comparing figure 1 with figure 5 it would appear as if the ITCZ and the SIOCZ are synonymous, but the text line 330 couples the SIOCZ to TTT, and line 334-336 clearly decouples the SIOCZ and the ITCZ. Certainly in the modern system the SIOCZ and TTT systems are distinct from the ITCZ. Since figure 1 includes the ITCZ, the SHW and the circulation patterns, it should also indicate the modern SIOCZ and TTT systems. The text that describes the contribution of ITCZ summer rainfall in the relevant catchment during glacials (lines 343-347) but in figure 5 the source of summer rainfall is indicated as the SIOCZ. This needs to be reconciled.

In the discussion of SST (line 237-240), comparison is between core top SST with modern SST data from Fallet et al. 2012 in order to defend a seasonal interpretation of the G. ruber Mg/Ca values. This argument is flawed in many ways. First, the uppermost Mg/Ca result from the sediment core is from 40.5cm depth in the core, which is approximately 5 000 years old according to the radiocarbon dates. This cannot be compared with the "modern" data from Fallet et al. (2012) which is approximately 1 000 years old. Indeed the age of the youngest Mg/Ca SST value prevents any verification against modern SST values. Second, the satellite data for SST in the Mozambique Channel presented by Fallet et al. (2012), and also the SST data based on Locarnini et al. (2013) presented in figure 1 show a strong thermal gradient in the Mozambique Channel. Correlating the Mg/Ca 27°C SST temperature for the "top" of this sediment core does not take in to account this southward cooling gradient. The inshore location of this core in the Delagoa Bight also implies stronger coastal influences that is associated with warmer SST (also based on data in Fallet et al. 2012). The seasonality of this SST reconstruction is not central to the development of the climate forcing argument, but it will need to be tempered as a stand-alone interpretation.

Minor issues

The age model for the core is very clearly argued, and is sufficiently convincing for the broad-brush stroke assessment of palaeoenvironmental proxies, but close scrutiny of

the radiocarbon dates indicates some heterogeneity in deposition rates. Rapid deposition is indicated between 300cm and 200 cm, and also between 150cm and 100cm (although the Bacon model produces a parsimonious smoothing that downplays the date from 102cm). Placing more emphasis on the outlier date leads to the possibility of very slow deposition in the 15 000-6 000 year range, and also in the 32 000 -25 000 year range. The Bacon model needs more input data to verify this level of heterogeneity, and so there is possibly an underestimation of the error in the age model around these periods of slow deposition. Similarly only 2 $\delta$18O tie points are used in the chronology for the oldest 60 000 year part of the record. This clearly cannot capture heterogeneity in the deposition rate, and again the age model error estimates are probably too small.

The suite of proxies that reflect wet and dry conditions in the catchment are reported to change in concert with one another, and this is clear in a relative sense but not in an absolute sense. Scrutiny of figure 4, for example, shows clear oscillations in values that are syncronised between proxies, but within proxies these oscillations are really most apparent because of the contrasting peaks and trough values that are immediately older or younger. The absolute values do not hold up to the wet/dry assignations. The K/Al and Ca/Fe ratios in the wet period around 82 000 years ago, for example, have very similar values to the arid values at around 46 000 and 52 000 years ago, and so the absolute values are seemingly not important. Some discussion of the relative nature of these proxies should be presented.

The interpretation of the $\delta$13C record invokes a framework presented by Dupont et al. (2011) in which woodlands and forests with grasslands in the interior during interglacials is contrasted with rivers fringed with gallery forests & sedges in glacials. This scenario may account for the observed trends in the record, but it is a very imprecise science. The entire $\delta$13C variability noted in the 100 000 year record all falls very in the range of C3 plants, and even the maximum values that are interpreted as an increased C4 plant community still fall in the C3 range. As much as this represents an integrated

C3/C4 environmental shift, it could just as well represent a xeric/mesic environment with exactly the same C3 plant communities. The part of the equation that is poorly developed is the source of the C in the marine sediments: it represents differential assimilation as a function of river length, and differential assimilation of C as a function of organic gradients from a riverbank to the watersheds. It is possibly too much to anticipate that this level of interpretation can be assigned to the observed pattern

The role of sedges in the $\delta$13C record interpretation also needs closer consideration. Stock et al. (2004 Austral Ecology) suggest that 14% of sedges are C4 in winter rainfall areas and 67% are C4 in summer rainfall areas. Seasonality of rainfall is clearly a controlling factor in the C3/C4 pathways for sedges, but the interpretation of the sediment core $\delta$13C record seems to hint that they are all C4.

The association between the wet/dry cycles portrayed in the core, and Heinrich Events and the Antarctic Isotope Maxima events is important in resolving the underlying climate forcing. It should be noted that HS4 is the negative excursion in the NGRIP $\delta$18O record around 37 000 years ago (possibly older as it is portrayed in figure 4 – maybe 38 000 -40 000 years ago). It is associated with a dry interval (red shading in figure 4) but the text associates it with a wet period (lines 395-399). Overall the association between wet/dry phases in the core proxies and the AIM and HS data is dependent on the errors in the age model, which was argued to be underestimated, but still comprises several thousand years in the older portion of the core.

It would be useful for those who will undoubtedly make use of this record in their research if the supplementary tables include a model age assignation, and not just the sample depth in the core.

Figures and figure captions

Figure 1: Please depict the SIOCZ and TTT because it is relevant in the discussion. Wonderkrater is depicted in the wrong place (somewhere in Zimbabwe). In reality it is well within the Limpopo catchment. Figure 2: The caption mentions "LR04" twice

in a redundant manner. Figure 3: This caption needs to be rewritten. It is difficult to decipher what is being referred to because of a random sprinkling of right parentheses and colons. Figure 4: This caption attributes blue or green shading as wet, "while wet phases are marked in red or yellow". Presumably one of these is dry. What is described as blue appears purple – this may be a personal problem, but possibly re-consider the colour that is used. The text "related to low pressure cells" is correct but confusing in its detail and should be revised.

Typos

Line 66: winterly should be winter Line 67, 114-115, 330, 334-336: Define the SIOCZ, is this the same as TTT (in fig 5 it seem synonymous with the southern extent of the ITCZ, but line 330 couples it to TTT, and line 334-336 clearly decouples the SIOCZ and the ITCZ) and also put it on to fig 1 as it comes up repeatedly Line 76: Re introduces the SIOCZ acronym Line 201: permil, but on line 139 per mil. Please be consistent throughout the text Line 244: Fig. 1a should be Fig. 3a

―――――――――――――――――――

---

## Referee Comment (RC2) · Anonymous Referee #2 · 24 Sep 2020

Based on a sediment core from Delagoa Bight offshore southeastern Africa, Hahn and co-authors present a new multi-proxy reconstruction of the continental climate for the last 100,000 years. The new record has high potential to improve our understanding how continental wetness has varied in response to latitudinal shifts in the westerlies and South Indian Ocean convergence zone. The data are certainly of very good quality and the new record has great potential, which, however, is not fully exploited in the current version of the manuscript. In my view there are several major shortcomings (see comments below) and major revisions are therefore required before the manuscript can be accepted for publication in CoP. I would like to emphasize that I will focus only on major issues at this stage of the review process: • The study site appears to be ideally situated to record displacements of the westerlies and the South Indian Con-

vergence Zone. Unfortunately, the authors do not really present a more detailed figure of the present-day atmospheric circulation patterns, which would help the readers to understand the discussion better. Basically, more detailed information on the atmospheric dynamics and according figures are required, such as the one presented by Charlotte Miller an co-authors in a previously published article in Climate of the Past (Figure 1 in Miller, C., et al. (2019). "Late Quaternary climate variability at Mfabeni peatland, eastern South Africa." Climate of the Past 15(3): 1153-1170. • Although multiple proxies were measured, there is rather little and very rudimentary information on their paleoclimatic significance and potential uncertainties and limitations are not discussed. For instance, the precipitation indicators $\delta$D, K/Al, Ca/Fe and red/blue ratios are only very briefly presented in paragraph 3.1.3. All proxies depend to varying extents on precipitation, erosion and fluvial transport, whereas these factors do not necessarily vary in concert. For instance, erosion is not always directly linked to the amount of precipitation and vegetation density is often an additional and more important factor for erosion rates. Erosion rates can also increase substantially at times of rapid climatic and associated vegetation changes. Because the relationship between precipitation and erosion (and riverine transport) is not linear. I would like to see a more critical discussion about the strength and weaknesses of the proxies. • Some of the authors have worked for a long time in this region and published multiple articles on past climate variability in this region. It is therefore quite surprising that there are no attempts to incorporate other continental records from South Africa more effectively into this study. Some of the records are mentioned in the text but not displayed in a figure. • The major precipitation indicators are presented in Figure 4, together with ice core records from both poles. The authors try to mark wet periods associated with different atmospheric circulation regimes. However, it remains absolutely enigmatic which scientific criteria were actually used to determine these periods. The width of the color-coded bars seems to be rather arbitrary as, for instance, indicated by the width of the green bar during MIS 5, which do not really match the minima in the $\delta$D and K/Al records. The authors must explain in close detail which criteria were used to

determine the different climatic phases. Furthermore, what is actually happening during the white intervals?  c Figure 5 is a basic conceptual model, but it also highlights the problem of this study as other records were not really used to support this basic model. The authors suggest that the major changes on glacial interglacial time scales are related to latitudinal shifts of atmospheric boundaries and westerlies. Are there no zonal shifts in the moisture transport? Furthermore, I would like to see a third figure showing the conceptual model for the present-day situation.

---

## Author Comment (AC2) · 9 Oct 2020

REBUTTAL COMMENTS IN CAPITAL LETTERS
tial, which, however, is not fully exploited in the current version of the manuscript. In my view there are several major shortcomings (see comments below) and major revisions are therefore required before the manuscript can be accepted for publication in CoP. I would like to emphasize that I will focus only on major issues at this stage of the review process: WE WOULD LIKE TO THANK THIS REVIEWER FOR THEIR CONSTRUC-TIVE COMMENTS AND WE HAVE REPLIED TO EACH COMMENT INDIVIDUALLY BELOW. THE REVIEWER COMMENTS ARE IN BLACK AND OUR RESPONSES IN BLUE. âAËŸ c The study site appears to be ′ ideally situated to record displacements of the westerlies and the South Indian convergence Zone. Unfortunately, the authors do not really present a more detailed figure of the present-day atmospheric circulation patterns, which would help the readers to understand the discussion better. Basically, more detailed information on the atmospheric dynamics and according figures are required, such as the one presented by Charlotte Miller an co-authors in a previously published article in Climate of the Past (Figure 1 in Miller, C., et al. (2019). "Late Quaternary climate variability at Mfabeni peatland, eastern South Africa." Climate of the Past 15(3): 1153-1170. WE HAVE ADDED SUB FIGURES 1 B AND 1C AS WELL AS THE FOLLOWING MORE DETAILED DESCRIPTION OF THE REGIONAL ATMOSPHERIC DYNAMICS IN THE "REGIONAL SETTINGS "SECTION (LINES 114-128).:" ALTHOUGH THE ITCZ CURRENTLY DOES NOT DIRECTLY AFFECT THE REGION, IT DOES INDUCE LATITUDINAL SHIFTS IN THE SIOCZ, WHICH CAN BE CONSIDERED AS A SOUTHWARD EXTENSION OF THE ITCZ. WHEN THE ITCZ IS IN ITS SOUTHERNMOST (SUMMER) POSITION, TROPICAL TEMPER-ATE TROUGHS (TTTS), FORMING AT THE SIOCZ BRING EASTERLY RAINFALL FROM THE INDIAN OCEAN (JURY ET AL., 1993; REASON AND MULENGA, 1999) (FIG 1B). DURING AUSTRAL SUMMER, A LOW-PRESSURE CELL DOMINATES THE SOUTHERN AFRICAN INTERIOR, ENABLING TROPICAL EASTERLIES/TTT TO BRING RAINFALL TO THE REGION. THIS RAINFALL IS SUPPRESSED DUR-ING AUSTRAL WINTER, WHEN A SUBTROPICAL HIGH-PRESSURE CELL IS LO-CATED OVER SOUTHERN AFRICA, (FIG. 1B). THIS HIGH-PRESSURE CELL CRE-

ATES A BLOCKING EFFECT OVER THE CONTINENT, WHICH STOPS MOISTURE ADVECTION INLAND OVER THE MAJORITY OF SOUTH AFRICA DURING WINTER (DEDEKIND ET AL., 2016). THE WINTER RAIN THAT DOES FALL (33 % OF ANNUAL RAINFALL FROM APRIL TO OCTOBER) IS ASSOCIATED WITH EXTRA-TROPICAL CLOUD BANDS AND THUNDERSTORMS LINKED TO FRONTAL SYSTEMS THAT DEVELOP IN THE MAIN SHW FLOW (BETWEEN 40 °S AND 50 °S). AS THE SHW SHIFT NORTHWARD DURING THE WINTER, THESE FRONTAL SYSTEMS MAY BECOME CUT OFF AND DISPLACED EQUATORWARD AS FAR NORTH AS 25°S (C.F. BARAY ET AL., 2003; MASON AND JURY, 1997) (FIG 1C). " âAËŸ c Although ′ multiple proxies were measured, there is rather little and very rudimentary information on their paleoclimatic significance and potential uncertainties and limitations are not discussed. For instance, the precipitation indicators $\delta$D, K/Al, Ca/Fe and red/blue ratios are only very briefly presented in paragraph 3.1.3. All proxies depend to varying extents on precipitation, erosion and fluvial transport, whereas these factors do not necessarily vary in concert. For instance, erosion is not always directly linked to the amount of precipitation and vegetation density is often an additional and more important factor for erosion rates. Erosion rates can also increase substantially at times of rapid climatic and associated vegetation changes. Because the relationship between precipitation and erosion (and riverine transport) is not linear. I would like to see a more critical discussion about the strength and weaknesses of the proxies. DD IS INDEED OUR ONLY "REAL" PRECIPITATION INDICATOR WHEREAS THE REMAINING PROXIES REFLECT EROSION, FLUVIAL TRANSPORT AND THE WEATHERING OF THE TRANSPORTED MATERIAL. ALL OF WHICH ARE INDEED LIABLE TO HAVE A NON-LINEAR RELATIONSHIP WITH PRECIPITATION AMOUNT. HOWEVER, SEEING THAT THE FOUR PROXIES (MOSTLY) CORRELATE IN OUR RECORD, THIS DOES NOT SEEM TO BE THE CASE FOR THE MOST PART OF OUR RECORD. WE HAVE ADDED THE FOLLOWING TEXT TO THE PARAGRAPH IN QUESTION (3.1.3) LINES 317-329: "" WE ALSO NOTE THAT OF THE FOUR PROXY INDICATORS ($\Delta$DC31, RED/BLUE, K/AL AND CA/FE) ONLY $\Delta$DC31 CAN

BE CONSIDERED AS DIRECT INDICATOR OF PAST PRECIPITATION CHANGE. RED/BLUE, K/AL AND CA/FE DEPEND TO VARYING EXTENTS ON PRECIPITATION, EROSION AND FLUVIAL TRANSPORT, WHEREAS THESE FACTORS DO NOT NECESSARILY VARY IN CONCERT. FOR INSTANCE, EROSION IS NOT ALWAYS DIRECTLY LINKED TO THE AMOUNT OF PRECIPITATION AND VEGETATION DENSITY IS OFTEN AN ADDITIONAL AND MORE IMPORTANT FACTOR FOR EROSION RATES. EROSION RATES CAN ALSO INCREASE SUBSTANTIALLY AT TIMES OF RAPID CLIMATIC AND ASSOCIATED VEGETATION CHANGES. BECAUSE THE RELATIONSHIP BETWEEN PRECIPITATION, EROSION AND RIVERINE TRANSPORT IS NOT LINEAR WE BASE OUR PRECIPITATION RECONSTRUCTION (I.E. THE DEFINITION OF THE ARID AND WET INTERVALS DESCRIBED IN SECTION 3.2 AND COLORED-CODED IN FIG. 4) MAINLY ON THE △DC31 VALUES. WE CONSIDER THE RED/BLUE , K/AL AND CA/FE VALUES AS SUPPORTIVE INFORMATION; THE RELATIVE CORRELATION OF THE FOUR PROXIES SUGGESTS THAT PHASES OF INCREASED PRECIPITATION ARE, FOR THE MOST PART, ASSOCIATED WITH AN INCREASE IN EROSION RATES, CHEMICAL WEATHERING AND RIVERINE TRANSPORT. THIS UNDERLINES THE RELIABILITY OF OUR PALEO-PRECIPITATION RECONSTRUCTION. "

âAËŸ c Some ′ of the authors have worked for a long time in this region and published multiple articles on past climate variability in this region. It is therefore quite surprising that there are no attempts to incorporate other continental records from South Africa more effectively into this study. Some of the records are mentioned in the text but not displayed in a figure. WE ARE UNSURE AS TO WHICH RECORDS THE REVIEWER IS REFERRING TO. RECORDS THAT SPAN THE TIME FRAME IN QUESTION AND AT THE SAME TIME HAVE A RESOLUTION THAT IS COMPARABLE TO THAT OF CORE GEOB20616 ARE VERY RARE IN THE REGION. THE ONLY AVAILABLE RECORDS ARE LOCATED MUCH FURTHER NORTH AND THUS OUT OF THE INFLUENCE OF THE CLIMATIC SYSTEMS WE ARE DESCRIBING. âAËŸ c The major precipitation indicators are presented in Figure 4, together with ′ ice core records from

both poles. The authors try to mark wet periods associated with different atmospheric circulation regimes. However, it remains absolutely enigmatic which scientific criteria were actually used to determine these periods. The width of the color-coded bars seems to be rather arbitrary as, for instance, indicated by the width of the green bar during MIS 5, which do not really match the minima in the $\delta$D and K/Al records. The authors must explain in close detail which criteria were used to determine the different climatic phases. Furthermore, what is actually happening during the white intervals? WE HAVE DETAILED THAT THE DEFINITION OF THE DIFFERENT CLIMATIC PHASES IS MAINLY BASED ON THE DD VALUES, THIS IS OUR MOST DIRECT PRECIPITATION PROXY AND THE RED/BLUE RATIOS AS WELL AS ELEMENTAL RATIOS SERVE MAINLY AS SUPPORTIVE INFORMATION, UNDERLINING THE RELIABILITY OF OUR PALEO-RAINFALL RECONSTRUCTION. WE DEFINITELY NEEDED TO CLARIFY THIS AND HAVE ADDED THE FOLLOWING TEXT TO LINES 323-328: "LINEAR WE BASE OUR PRECIPITATION RECONSTRUCTION (I.E. THE DEFINITION OF THE ARID AND WET INTERVALS DESCRIBED IN SECTION 3.2 AND COLORED-CODED IN FIG. 4) MAINLY ON THE $\triangle$DC31 VALUES. WE CONSIDER THE RED/BLUE , K/AL AND CA/FE VALUES AS SUPPORTIVE INFOR-MATION; THE RELATIVE CORRELATION OF THE FOUR PROXIES SUGGESTS THAT PHASES OF INCREASED PRECIPITATION ARE, FOR THE MOST PART, ASSOCIATED WITH AN INCREASE IN EROSION RATES, CHEMICAL WEATH-ERING AND RIVERINE TRANSPORT. THIS UNDERLINES THE RELIABILITY OF OUR PALEO-PRECIPITATION RECONSTRUCTION ." CONCERNING THE WHITE PHASES; THESE WE CONSIDER AS TRANSITIONAL PERIODS, AS IS NOW MARKED IN THE CAPTION OF FIG 4. âAËŸ c Figure 5 is a basic conceptual model, but it also highlights ′ the problem of this study as other records were not really used to support this basic model. WE UNDERSTAND THAT THE REVIEWER WOULD LIKE TO SEE A MORE THOROUGH COMPARISON OF OUR RECORD WITH OTHER REGIONAL CONTINENTAL RECORDS. HOWEVER, THERE ARE FEW/NONE RECORDS THAT SPAN THE TIME FRAME IN QUESTION AND AT THE SAME TIME

HAVE A RESOLUTION THAT IS COMPARABLE TO THAT OF CORE GEOB20616. THE ONLY AVAILABLE RECORDS ARE LOCATED MUCH FURTHER NORTH AND THUS OUT OF THE INFLUENCE OF THE CLIMATIC SYSTEMS WE ARE DESCRIBING. The authors suggest that the major changes on glacial interglacial time scales are related to latitudinal shifts of atmospheric boundaries and westerlies. Are there no zonal shifts in the moisture transport? THIS IS AN INTERESTING POINT; HOWEVER WE FIND NO EVIDENCE FOR ZONAL SHIFTS IN THE MOISTURE TRANSPORT. THERE IS A DIVIDE (CAB, CONGO AIR BOUNDARY) BETWEEN ATLANTIC AND INDIAN OCEAN MOISTURE BUT IT IS LOCATED VERY CLOSE TO THE ATLANTIC COAST (THE ATLANTIC MOISTURE SIMPLY DOES NOT MAKE IT TO THE INTERIOR DUE TO THE BENGUELA UPWELLING). ONLY UNDER CONDITIONS WITHOUT BENGUELA UPWELLING (I.E. BEFORE THE MIOCENE ESSENTIALLY) IT WOULD HAVE BEEN POSSIBLE THAT THE CAB WAS LOCATED FURTHER EAST AND ATLANTIC MOISTURE WOULD MAKE IT TO THE EASTERN COAST OF SA. UNDER THE MODERN CLIMATE (UPWELLING, ATMOSPHERE) SYSTEM, EVEN UNDER GLACIAL STATE, IT IS SIMPLY NOT POSSIBLE. Furthermore, I would like to see a third figure showing the conceptual model for the present-day situation. THE PRESENT DAY SITUATION WOULD CORRESPOND TO THE "INTERGLACIAL STATE". WE HAVE MARKED THIS ACCORDINGLY IN THE CAPTIO

Please also note the supplement to this comment:
https://cp.copernicus.org/preprints/cp-2019-158/cp-2019-158-AC2-supplement.pdf

[Figure]

**Fig. 1.** revised fig 1

[Figure]

[Figure]

★ site GeoB20616-1

**Fig. 2.** revised fig 5

**Supplement:**

[revised manuscript text omitted]

---

## Author Response (AR1)

[revised manuscript text omitted]

**First reviewer comments and rebuttal**

Summary

In this article, Hahn et al. report on the analysis of a 958 cm sediment core that was taken in the Delagoa
Bight off southeastern Africa. The source of the sediment is argued to be from three nearby river
catchments that are relatively small, and as a result of this the environmental information derived from
the sediments represents a fairly clean signal (incontrast to nearby cores that sample the Limpopo River
catchment which is vast and probably includes multiple climate sensitivities). The chronology for the core
is generated using 12 radiocarbon dates, of which two are beyond the limit of the method, and δ18O
values from benthic foraminifera that are compared to the LR04 benthic stack. The radiocarbon dates from
the upper part of the record overlap with the δ18O values from the lower part of the stack so that one of
the tie δ18O tie points has an apriori age assignment. The results of the age model demonstrate a relatively
constant deposition over the last 100 000 years. Ca/Mg ratios on foraminifera are used to reconstruct past
sea surface temperature (SST), and this record demonstrates that SST was almost 4◦C warmer during
interglacials (MIS 5 & 1) that it was during glacials (MIS4-2). The main substance of the article is the
presentation of multiple geochemical tracers of terrestrial climate and the associated vegetation
responses in the catchment. These include δ13C C31 (terrestrial plants community indicator), red/blue
ratios (organic vs. classic indicators), K/Al (chemical vs. physical weathering), Ca/Fe (terrestrial vs. marine
source indicator) and δDC31 values (rainfall amount indicators). The authors identify a coherent pattern
in which all of the geochemical tracers vary in concert with one another, and this is coherently argued to
reflect hydrological changes in the associated river catchments. The underlying cause for the alternation
between mesic and xeric conditions in the catchment are explored through northern hemisphere forcing
in the form of Heinrich Stadials, and southern hemisphere forcing in the form of Antarctic ice advances.
What emerges is that the forcing during glacials and interglacials differ from one another, and this must be reconciled through synoptic scale changes in the drivers of continental rainfall (rather than insolation
variability). The model that is proposed centers on the way in which the two main moisture-bearing
systems, the inter-tropical convergence zone (ITCZ) and the southern hemisphere westerlies (SHW),
interact and influence the development of the South African high system that is dominant influence on
modern rainfall. In essence the argument is that during glacial conditions the SHW migrate northwards
because of Antarctic expansion, while the thermal equator (ITCZ) migrates southward because of Arctic
expansion, and the South African high system occludes. The extinction of the South African high pressure
system during glacials prevents the development of the southern Indian Ocean convergence zone (SIOCZ)
and the associated temperate tropical troughs (TTT) that dominate modern summer rainfall (but please
note the comment on this subject below). As a result, the catchment maintained a relatively constant
water balance between glacials and interglacials as the glacial loss of the TTT/SIOCZ was compensated by
direct summer rainfall from the ITCZ and/or winter rainfall from the southern hemisphere westerlies
(depending on the dominant Arctic vs. Antarctic forcing at the time).

Scientific merit

Notwithstanding the critique that is presented below, this manuscript makes a valuable contribution to
climate science in southern Africa. The dynamics of the climate system are relevant to both future projects
of climate change, and the interpretation of the rich archaeological heritage of the region. Several
archaeological sequences of a similar age are in close enough proximity for the climate model to be
relevant. Sibudu cave and Border Cave contain evidence of mesic/xeric cycles, and they are also well dated,
so there is potential to refine the glacial/interglacial climate model. Key palaeoclimate records, such as the
Pretoria Salt Pan have been dated using insolation arguments, and if the climate model proposed by Hahn
et al. is correct, then the basis for the age model of the Pretoria Salt Pan is flawed. The authors have
presented their data in supplementary tables, which is going to be very useful for comparing this dataset
to others.

We would like to thank this reviewer for their constructive comments and we have replied to each
comment individually below. The reviewer comments are in black and our responses in blue.

Review

The critique of the manuscript is in the form of substantive clarifications, minor issues, and typos.

Substantive clarification

One of the most important contributions of this manuscript is the model for synoptic shifts in the region
during glacial periods, and in particular its effect on the SIOCZ. Clarification is required of exactly what the
SIOCZ is. Comparing figure 1 with figure 5 it would appear as if the ITCZ and the SIOCZ are synonymous,
but the text line 330 couples the SIOCZ to TTT, and line 334-336 clearly decouples the SIOCZ and the ITCZ.
Certainly in the modern system the SIOCZ and TTT systems are distinct from the ITCZ. Since figure 1
includes the ITCZ, the SHW and the circulation patterns, it should also indicate the modern SIOCZ and TTT
systems.

Yes, the relationship between SIOCZ and ITCZ needs to be better explained. We have added the modern
SIOCZ and TTT systems to figure 1. Furthermore, we have detailed the relationship between the ITCZ and
the SIOCZ in the text (section: "regional setting" lines 113-116: ). Following the request by reviewer 2 we
have also detailed the regional climate section a bit more also adding fig 1b and c.

The text that describes the contribution of ITCZ summer rainfall in the relevant catchment during glacials
(lines 343-347) but in figure 5 the source of summer rainfall is indicated as the SIOCZ. This needs to be
reconciled.

Yes; this an error in Figure 5 that has been corrected now. What was initially marked as "SIOCZ" was actually the ITCZ, this has been changed and the SIOCZ position is now indicated as well. Concerning the text on glacial periods: the southward shifts of the ITCZ (due to a more southerly position of the thermal equator) would entail southward shifts of the SIOCZ. It is unclear if this, or a direct ITCZ influence caused humid phases during Heinrich events. This has been added to the section "3.2.1.2. Hydrology over glacial-interglacial transitions" lines 359-363.

In the discussion of SST (line 237-240), comparison is between core top SST with modern SST data from Fallet et al. 2012 in order to defend a seasonal interpretation of the G. ruber Mg/Ca values. This argument is flawed in many ways. First, the uppermost Mg/Ca result from the sediment core is from 40.5cm depth in the core, which is approximately 5 000 years old according to the radiocarbon dates. This cannot be compared with the "modern" data from Fallet et al. (2012) which is approximately 1000 years old. Indeed the age of the youngest Mg/Ca SST value prevents any verification against modern SST values.

Second, the satellite data for SST in the Mozambique Channel presented by Fallet et al. (2012), and also the SST data based on Locarnini et al. (2013) presented in figure 1 show a strong thermal gradient in the Mozambique Channel. Correlating the Mg/Ca 27°C SST temperature for the "top" of this sediment core does not take in to account this southward cooling gradient. The inshore location of this core in the Delagoa Bight also implies stronger coastal influences that is associated with warmer SST (also based on data in Fallet et al. 2012). The seasonality of this SST reconstruction is not central to the development of the climate forcing argument, but it will need to be tempered as a stand-alone interpretation.

This is true; we have removed the comparison with the core top sample. We have left the reference to Wang et al., (2013) saying that the authors "suggest that $U^{K'}_{37}$ SST reflects warm season SST mediated by changes in the Atlantic, whereas the *G. ruber* Mg/Ca SST indicator used in this study records cold season SST mediated by climate changes in the southern hemisphere.". Although we agree with the reviewer that the seasonality of the SSTs is not central to our argumentation, but we thought we would leave this suggestion as information for the reader.

Minor issues

The age model for the core is very clearly argued, and is sufficiently convincing for the broad-brush stroke assessment of palaeoenvironmental proxies, but close scrutiny of the radiocarbon dates indicates some heterogeneity in deposition rates. Rapid deposition is indicated between 300cm and 200 cm, and also between 150cm and 100cm (although the Bacon model produces a parsimonious smoothing that downplays the date from 102cm). Placing more emphasis on the outlier date leads to the possibility of very slow deposition in the 15 000-6 000 year range, and also in the 32 000 -25 000 year range. The Bacon model needs more input data to verify this level of heterogeneity, and so there is possibly an underestimation of the error in the age model around these periods of slow deposition. Similarly only 2 δ18O tie points are used in the chronology for the oldest 60 000 year part of the record. This clearly cannot capture heterogeneity in the deposition rate, and again the age model error estimates are probably too small.

We agree with the reviewer and have added this to the age model error estimation in section "2.3. age model" See lines 156-171

The suite of proxies that reflect wet and dry conditions in the catchment are reported to change in concert with one another, and this is clear in a relative sense but not in an absolute sense. Scrutiny of figure4, for example, shows clear oscillations in values that are synchronized between proxies, but within proxies these oscillations are really most apparent because of the contrasting peaks and trough values that are immediately older or younger. The absolute values do not hold up to the wet/dry assignations. The K/Al
and Ca/Fe ratios in the wet period around 82 000 years ago, for example, have very similar values to the
arid values at around 46 000 and 52 000 years ago, and so the absolute values are seemingly not important.
Some discussion of the relative nature of these proxies should be presented.

We agree and have added this idea to the end of the "3.1.3 Precipitation indicators" section lines 313-
328

The interpretation of the δ13C record invokes a framework presented by Dupont et al. (2011) in which
woodlands and forests with grasslands in the interior during interglacials is contrasted with rivers fringed
with gallery forests & sedges in glacials. This scenario may account for the observed trends in the record,
but it is a very imprecise science. The entire 13Cvariability noted in the 100000year record all falls very in
the range of C3 plants, and even the maximum values that are interpreted as an increased C4 plant
community still fall in the C3 range. As much as this represents an integrated C3/C4 environmental shift,
it could just as well represent a xeric/mesic environment with exactly the same C3 plant communities.

The entire 13C variability noted in the 100000year record (-26 to -23‰) is indeed relatively small and close
to the 13C values expected for alkanes from C4 plants (around -20‰, depleted up to -25‰; compare
dataset 'all Africa for C31 alkane in Garcin et al., 2014). However, there are only few C4 plants with 13C
values as depleted as -25‰ so the variability observed in our record must be caused by variable
contributions from C3 plants. Other indications for this:

A)  if shifts in d13C were only dependent on shifts in hydrology (xeric to mesic) then there should be
a correlation between the d13C and dD variability (dD is a reliable indicator of rainfall amount).
Since this is not the case ($R^2$=0.15), we assume that a further factor (i.e. shift in vegetation biomes)
drives d13C values.
B)  In order to definitely answer the question on what is driving d13C values (vegetation change vs
hydrological change) we would need pollen data for our core. These would give us reliable infor-
mation on the actual vegetation changes in the catchment. Unfortunately, palynological analysis
will not be done at the site so that we have to refer to closest neighboring palynological dataset.
The core studied by Dupont et al. 2011 is in the direct vicinity of our site; and the catchments from
which the material is sourced are adjacent. Although the comparison of our record with the
Dupont et al. 2011 pollen data is imprecise, as the reviewer correctly remarks, it is the best we
can currently do. In section "3.2.1.1. Sea surface temperatures and vegetation" we describe in
detail ow our catchment areas differ from that of the Limpopo and the consequences this has on
vegetation signals. We find that in order to answer the question concerning the drivers of d13C
values (vegetation change vs hydrological change) the comparison with the Dupont et al. 2011
record is helpful in the sense that Dupont et al. 2011 document large changes in the vegetation
biomes of the Limpopo catchment over glacial-interglacial transitions. If this was the case in the
Limpopo catchment it is likely that similarly shifts in vegetation biomes took place in the adjacent
catchments.

We have tried to bring these arguments forth more clearly in the rewritten version of the section
lines250-285 Finally we note that we may not be able to pinpoint exactly what caused shifts in the
d13C record, but either mechanism would translate to heavier values during more arid conditions and
lighter values during wet conditions. We have tried to underline this in the discussion of the proxies
(lines 250-285).

The role of sedges in the δ13C record interpretation also needs closer consideration. Stock et al.
(2004AustralEcology) suggest that 14% of sedges are C4 in winter rainfall areas and 67% are C4 in summer rainfall areas. Seasonality of rainfall is clearly a controlling factor in the C3/C4 pathways for sedges, but the interpretation of the sediment core δ13C record seems to hint that they are all C4.

This is an interesting point and it is possible that variations in winter and summer rainfall may affect the vegetation and thus our d13C record. However, without downcore palynology with phytolith analysis to distinguish between C4 and C3 grasses, it is not possible to go into such detail. Looking at the data, it also seems unlikely that such an effect took place: if the shifts from winter to summer rain were driving d13C values, our data would suggest that glacial periods have increased summer rains, whereas interglacials had a more winter rainfall regime. This is highly unlikely; we know that modern (interglacial) climate is a summer rainfall regime and that only during glacials has the winter rainfall zone been inferred to shift northward. This interpretation of the d13C signal would therefore be very difficult to consolidate with what we know about the regional climate.

The association between the wet/dry cycles portrayed in the core, and Heinrich Events and the Antarctic Isotope Maxima events is important in resolving the underlying climate forcing. It should be noted that HS4 is the negative excursion in the NGRIP δ18O record around 37 000 years ago (possibly older as it is portrayed in figure 4 – maybe 38 000 -40 000 years ago). It is associated with a dry interval (red shading in figure 4) but the text associates it with a wet period (lines 395-399). Overall the association between wet/dry phases in the core proxies and the AIM and HS data is dependent on the errors in the age model, which was argued to be underestimated, but still comprises several thousand years in the older portion of the core.

Unfortunately, this is true, and we have pointed this out in the manuscript at the end of the section "*3.2.2.2. During MIS 4-2 glacial conditions*" discussion: lines 438-441

It would be useful for those who will undoubtedly make use of this record in their research if the supplementary tables include a model age assignation, and not just the sample depth in the core.

Good idea. This has been done.

Figures and figure captions Figure 1: Please depict the SIOCZ and TTT because it is relevant in the discussion.

Upon suggestion by the 2nd reviewer, We have added a new sub figure to fig1 that now shows in detail the modern climate system including SIOCZ and TTT...

Wonderkrater is depicted in the wrong place (somewhere in Zimbabwe). In reality it is well within the Limpopo catchment.

Yes! Changed...

Figure 2: The caption mentions "LR04" twice in a redundant manner.

Done

Figure 3: This caption needs to be rewritten. It is difficult to decipher what is being referred to because of a random sprinkling of right parentheses and colons.

We have rewritten the caption for better understanding

Figure 4: This caption attributes blue or green shading as wet, "while wet phases are marked in red or yellow". Presumably one of these is dry.

Yes, this has been corrected, the latter is dry.

What is described as blue appears purple – this may be a personal problem, but possibly re-consider the
colour that is used.

we have opted for a pure blue now…

The text "related to low pressure cells" is correct but confusing in its detail and should be revised.

Yes, this detail is confusing in the text, we have removed it.

Typos Line 66: winterly should be winter ok

Line 67, 114-115, 330, 334-336: Define the SIOCZ, is this the same as TTT (in fig 5 it seem synonymous
with the southern extent of the ITCZ, but line 330 couples it to TTT, and line 334-336 clearly decouples the
SIOCZ and the ITCZ)

Yes, the relationship between SIOCZ, TTT and ITCZ needs to be better explained. The SIOCZ is a southward
extension of the ITCZ. The tropical temperate troughs (TTTs) form at the SIOCZ. We have added the modern
ITCZ, SIOCZ and TTT systems to new subfigure of figure 1. Furthermore, we have detailed the relationship
between the ITCZ and the SIOCZ and TTT in the text (section: "regional setting" lines 113-116 ).

and also put it on to fig 1 as it comes up repeatedly ok

Line 76: Re introduces the SIOCZ acronym

This has been removed….

Line 201: permil, but on line 139 per mil. Please be consistent throughout the text ok

Line 244: Fig. 1a should be Fig. 3a

Yes!

**Second Reviewer Comments and Rebutall**

Based on a sediment core from Delagoa Bight offshore southeastern Africa, Hahn and co-authors present
a new multi-proxy reconstruction of the continental climate for the last 100,000 years. The new record has
high potential to improve our understanding how continental wetness has varied in response to latitudinal
shifts in the westerlies and South Indian Ocean convergence zone. The data are certainly of very good
quality and the new record has great potential, which, however, is not fully exploited in the current version
of the manuscript. In my view there are several major shortcomings (see comments below) and major
revisions are therefore required before the manuscript can be accepted for publication in CoP. I would like
to emphasize that I will focus only on major issues at this stage of the review process:

We would like to thank this reviewer for their constructive comments and we have replied to each
comment individually below. The reviewer comments are in black and our responses in blue.

âǍ˘ c The study site appears to be ´ ideally situated to record displacements of the westerlies and the
South Indian convergence Zone. Unfortunately, the authors do not really present a more detailed figure
of the present-day atmospheric circulation patterns, which would help the readers to understand the
discussion better. Basically, more detailed information on the atmospheric dynamics and according figures are required, such as the one presented by Charlotte Miller an co-authors in a previously published article
in Climate of the Past (Figure 1 in Miller, C., et al. (2019). "Late Quaternary climate variability at Mfabeni
peatland, eastern South Africa." Climate of the Past 15(3): 1153-1170.

We have added sub figures 1 b and 1c as well as a more detailed description of the regional atmospheric
dynamics in the "regional settings "section (lines 114-128).

âAˇ c Although ´ multiple proxies were measured, there is rather little and very rudimentary information
on their paleoclimatic significance and potential uncertainties and limitations are not discussed. For
instance, the precipitation indicators $\delta D$, K/Al, Ca/Fe and red/blue ratios are only very briefly presented in
paragraph 3.1.3. All proxies depend to varying extents on precipitation, erosion and fluvial transport,
whereas these factors do not necessarily vary in concert. For instance, erosion is not always directly linked
to the amount of precipitation and vegetation density is often an additional and more important factor
for erosion rates. Erosion rates can also increase substantially at times of rapid climatic and associated
vegetation changes. Because the relationship between precipitation and erosion (and riverine transport)
is not linear. I would like to see a more critical discussion about the strength and weaknesses of the proxies.

dD is indeed our only "real" precipitation indicator whereas the remaining proxies reflect erosion, fluvial
transport and the weathering of the transported material. All of which are indeed liable to have a non-
linear relationship with precipitation amount. However, seeing that the four proxies (mostly) correlate in
our record, this does not seem to be the case for the most part of our record. We have added these
considerations to the paragraph in question (3.1.3). Lines 317-329

âAˇ c Some ´ of the authors have worked for a long time in this region and published multiple articles on
past climate variability in this region. It is therefore quite surprising that there are no attempts to
incorporate other continental records from South Africa more effectively into this study. Some of the
records are mentioned in the text but not displayed in a figure.

We are unsure as to which records the reviewer is referring to. Records that span the time frame in
question and at the same time have a resolution that is comparable to that of core GeoB20616 are very
rare in the region. The only available records are located much further north and thus out of the influence
of the climatic systems we are describing.

âAˇ c The major precipitation indicators are presented in Figure 4, together with ´ ice core records from
both poles. The authors try to mark wet periods associated with different atmospheric circulation regimes.
However, it remains absolutely enigmatic which scientific criteria were actually used to determine these
periods. The width of the color-coded bars seems to be rather arbitrary as, for instance, indicated by the
width of the green bar during MIS 5, which do not really match the minima in the $\delta D$ and K/Al records.
The authors must explain in close detail which criteria were used to determine the different climatic
phases. Furthermore, what is actually happening during the white intervals?

We have detailed that the definition of the different climatic phases is mainly based on the dD values, this
is our most direct precipitation proxy and the red/blue ratios as well as elemental ratios serve mainly as
supportive information, underlining the reliability of our paleo-rainfall reconstruction. We definitely
needed to clarify this and have added an according section to lines 323-328.

Concerning the white phases; these we consider as transitional periods, as is now marked in the caption
of Fig 4.

âAˇ c Figure 5 is a basic conceptual model, but it also highlights ´ the problem of this study as other records
were not really used to support this basic model.

We understand that the reviewer would like to see a more thorough comparison of our record with other
regional continental records. However, there are few/none records that span the time frame in question
and at the same time have a resolution that is comparable to that of core GeoB20616. The only available
records are located much further north and thus out of the influence of the climatic systems we are
describing.

The authors suggest that the major changes on glacial interglacial time scales are related to latitudinal
shifts of atmospheric boundaries and westerlies. Are there no zonal shifts in the moisture transport?

This is an interesting point; however we find no evidence for zonal shifts in the moisture transport. There
is a divide (CAB, Congo Air boundary) between Atlantic and Indian Ocean moisture but it is located very
close to the Atlantic coast (the Atlantic moisture simply does not make it to the interior due to the
Benguela upwelling). Only under conditions without Benguela upwelling (i.e. before the Miocene
essentially) it would have been possible that the CAB was located further east and Atlantic moisture would
make it to the eastern coast of SA. Under the modern climate (upwelling, atmosphere) system, even under
glacial state, it is simply not possible.

Furthermore, I would like to see a third figure showing the conceptual model for the present-day situation.

The present day situation would correspond to the "interglacial state". We have marked this accordingly
in the caption.

-